# Precession-driven low-latitude hydrological cycle paced by shifting perihelion

Hu Yang[1], Xiaoxu Shi[1], Xulong Wang[2], Qingsong Liu[3], Yi Zhong[3], Xiaodong Liu[2], Youbin Sun[2], Yanjun Cai[4], Fei Liu[5], Gerrit Lohmann[6], Martin Werner[6], Zhimin Jian[7], Tainã M. L. Pinho[6], Hai Cheng[4], Lijuan Lu[5,1], Jiping Liu[5,1], Chao-Yuan Yang[1], Qinghua Yang[5,1], Yongyun Hu[8], Xing Cheng[9], Jingyu Zhang[3], and Dake Chen[1]

[1]Southern Marine Science and Engineering Guangdong Laboratory, Zhuhai, China
[2]State Key Laboratory of Loess and Quaternary Geology, Institute of Earth Environment, Chinese Academy of Sciences, Xi'an, China
[3]Centre for Marine Magnetism, Department of Ocean Science and Engineering, Southern University of Science and Technology, Shenzhen, China
[4]Institute of Global Environmental Change, Xi'an Jiaotong University, Xi'an, China
[5]School of Atmospheric Sciences, Sun Yat-sen University, Zhuhai, China
[6]Alfred Wegener Institute, Helmholtz Centre for Polar and Marine Research, Bremerhaven, Germany
[7]Laboratory of Marine Geology, Tongji University, Shanghai, China
[8]Department of Atmospheric and Oceanic Sciences, School of Physics, Peking University, Beijing, China
[9]ShanXi Experimental Center of Geological Survey, ShanXi Institute of Geological Survey, Xi'an, China

**Correspondence:** Hu Yang (yanghu@sml-zhuhai.cn)

**Abstract.** Paleoclimate proxies reveal a significant precessional impact on the low-latitude hydrological cycle. Classical theory suggests that precession modulates the inter-hemisphere summer insolation difference, and hence controls the meridional displacement of the Inter-Tropical Convergence Zone. Accordingly, low-latitude precipitation variations are expected to be in-phase (for the Northern Hemisphere) or anti-phase (for the Southern Hemisphere) with the Northern Hemisphere summer insolation. However, increasing number of proxies, particularly those absolutely dated ones, reveal that variations in terrestrial precipitation at different low-latitudes follow distinct precession rhythms that are very often out-of-phase with hemispheric summer insolation. The mechanism underlying such spatial–temporal complexity remains elusive. In this study, we performed theoretical analysis, climate simulations, and synthesis of geological records to hypothesize that the low-latitude hydrological cycle is paced by shifting perihelion rather than the hemispheric summer insolation. More specifically, precession of the Earth's rotation axis shifts the season and latitude of perihelion. Here, the latitude of perihelion is introduced as the latitude of Earth's subsolar point during perihelion, which is the location where the most intense solar radiation is concentrated. At the time of perihelion, intense solar radiation heats the land faster than the ocean due to differing thermal inertia. This thermodynamically moves the tropical convection from the ocean to the land, contributing to enhancing the terrestrial precipitation around the perihelion latitude. As the precessional phase changes, perihelion moves toward different latitudes, causing asynchronous maximums in terrestrial precipitation at different latitudes. Perihelion can occur in any season; therefore, the insolation in individual seasons is equally important in shaping the orbital-scale climate changes at low latitudes. This offers new insight into the Milankovitch theory, which highlights summer insolation's role in shaping orbital-scale climate change.

# 1 Introduction

By redistributing solar radiation across different latitudes and seasons, Earth's orbital changes exerted a significant impact on past climate change (Milankovitch, 1941; Berger, 1978, 1988). In high-latitude regions, these orbital-scale climate changes are characterized by cyclic expansion and retreat of ice sheets, with periodicities of ∼41 or ∼100 kilo-years (ka) (Lisiecki and Raymo, 2005), modulated by the Milankovitch cycles of obliquity and eccentricity (Milankovitch, 1941). In contrast, low-latitude climate changes are marked by periodic variations in hydrological cycle with dominant period of ∼23 ka, aligning with the Milankovitch cycles of precession (Clement et al., 2004; Cruz Jr et al., 2005; Braconnot et al., 2008; Carolin et al., 2013; Cheng et al., 2016; Wang et al., 2014a; Huang et al., 2020; Cheng et al., 2022). These precessional fluctuations are widely documented in the precipitation-related proxies from Africa (Kutzbach, 1981; Kuper and Kropelin, 2006; Hély et al., 2014; El-Shenawy et al., 2018), Asia (An et al., 2000; Wang et al., 2007, 2008; Carolin et al., 2013, 2016; Li et al., 2024), Australia (Beaufort et al., 2010) and South America (Baker et al., 2001; Wang et al., 2004; Cruz Jr et al., 2005). Among these changes, the periodic greening of the Sahara stands out as one of the most fascinating phenomena (DeMenocal et al., 2012; Skonieczny et al., 2019; Armstrong et al., 2023).

Low-latitude precipitation primarily comes from the seasonal north-south migration of the Inter-Tropical Convergence Zone (ITCZ), which follows the march of the Earth's thermal equator (Fig. 1) (Gadgil, 2018). Early studies proposed that precession regulates the inter-hemispheric summer insolation difference and modulates the meridional migration of the ITCZ (Kutzbach, 1981; Wang et al., 2014a; Schneider et al., 2014). Increased Northern Hemisphere summer insolation (perihelion) corresponds to decreased Southern Hemisphere summer insolation (aphelion). Accordingly, variations in ITCZ precipitation are expected to align with changes in summer insolation, exhibiting an anti-phase relationship between the Northern and Southern Hemispheres (Wang et al., 2004; Cruz Jr et al., 2005; Wang et al., 2006; Cheng et al., 2012, 2013; Wang et al., 2014a, 2017) (Fig. 2b and 2d). However, in the past two decades, an increasing number of proxies have shown that although low-latitude terrestrial precipitation displays significant precessional variations, their evolution patterns are not necessarily in-phase with the Northern/Southern Hemisphere summer insolation (An et al., 2000; Wang et al., 2004; Clemens and Prell, 2007; Carolin et al., 2013, 2016; Zhou et al., 2022). For example, the reconstructed optimums of the East Asian summer monsoon at different geographical locations were asynchronous with each other, suggesting that the precipitation optimum doesn't occur all at the same time (An et al., 2000; Cai et al., 2010; Ran and Feng, 2013; Chen et al., 2015; Liu et al., 2015; Zhou et al., 2022). The Indian summer monsoon, reconstructed from sediment cores of the Arabian Sea, shows an early Holocene optimum (Thamban et al., 2007), significantly preceding that of the East Asian summer monsoon (Sun et al., 2006; Liu et al., 2015). Over Africa, archaeological and geological evidence indicates that the recent termination of the greening Sahara shows time-transgressive pattern, occurred earlier in north Africa than in the south (Kuper and Kropelin, 2006; Shanahan et al., 2015). Speleothem records and travertine deposits collected around northeastern Brazil (10°S) suggested that the local wet periods presented themselves during the high insolation periods of austral autumn rather than austral summer (Wang et al., 2004). Near the equator, a Malaysian speleothem $\delta^{18}O$ record, reflecting local precipitation strength, exhibits an evolutionary pattern neither

comparable to the Northern Hemisphere summer insolation nor the Southern Hemisphere summer insolation, but insolation in October (Carolin et al., 2013) (Fig. 2c).

The cause of the asynchronous relationship between terrestrial precipitation and summer insolation has been investigated extensively form both modelling and proxy-model comparison perspectives (Short and Mengel, 1986; Braconnot and Marti, 2003; Zhao et al., 2005; Kuper and Kropelin, 2006; Clemens and Prell, 2007; Shanahan et al., 2015; Ran and Feng, 2013; Chen et al., 2015; Cheng et al., 2021; Zhou et al., 2022). These studies propose that summer insolation's control on the hydrological cycle could be disrupted by a variety of factors, including fluctuation of $CO_2$ (Lu et al., 2013; McGrath et al., 2021), internal ocean-atmosphere feedback (Clemens and Prell, 2007; Carolin et al., 2013, 2016), presence of ice sheet (Chiang and Bitz, 2005; Chen et al., 2015; Wu et al., 2023b), delayed ocean warming (Ran and Feng, 2013), sea level fluctuations (Griffiths et al., 2009; Windler et al., 2021) and millennial-scale abrupt climate changes originating from high latitudes (Wang et al., 2008; Chiang and Friedman, 2012). In addition to these possibilities, recent studies have shown that the sensitivity of different proxies to precipitation may lead to artificial "asynchronous" timing of reconstructed precipitation optimum (Cheng et al., 2021; Zheng et al., 2023; Wen et al., 2024).

Despite these facts, transient and time-slice simulations with solely insolation forcing yield an asynchronous precipitation response to hemispheric summer insolation (Braconnot and Marti, 2003; Kutzbach et al., 2008; Erb et al., 2015; Bischoff et al., 2017; Liu et al., 2022). This raises question of whether terrestrial precipitation follows (or not) changes in hemispheric summer insolation. By performing simulations with different precessional phases, Braconnot and Marti (2003) found that the calendar timing of maximum insolation (perihelion) affects the seasonality of Indian monsoon. They proposed that insolation in other seasons may also important, especially in determining the seasonal timing of Indian summer monsoon. Focusing on the East Asian monsoon, Zhou et al. (2022) suggested that insolation in different months may contribute to a time-transgressive pattern of East Asian monsoon optimums, with earlier occurrence in the southern China and later occurrence in the northern area. By running an isotope-enabled transient simulation covering the past 300 ka, Liu et al. (2022) found that the precipitation $\delta^{18}O$ in Asia is primarily affected by the temperature and insolation during the rainy seasons, rather than the averaged boreal summer insolation.

In this study, we present theoretical analysis, climate model simulations, and synthesis of geologic records from different latitudes to hypothesize that the precession-driven low-latitude hydrological cycle is regulated by shifting perihelion across different seasons and latitudes. Consequently, the precipitation optimums at different latitudes occurs in a naturally asynchronous manner.

## 2 Methodology

### 2.1 Model Simulations

Using Alfred Wegener Institute Earth System Model (AWI-ESM, (Sidorenko et al., 2015, 2019)), we conduct two set of experiments to study the dynamics of precession-driven low-latitude hydrological cycle. The AWI-ESM is a coupled ocean-atmosphere model, consisting the atmospheric component ECHAM6 (Giorgetta et al., 2013) and the oceanic component FE-

SOM (Wang et al., 2014b; Danilov et al., 2017). The oceanic component employs an unstructured-mesh, with relatively high resolution (up to 25 km) at polar regions, coast areas and the equator, while the atmospheric component has a spatial resolution of 1.875°. In our experiments, we implement a dynamic vegetation, which dynamically alters the vegetation coverage and the surface albedo in response to climate changes. More detailed information on the model setup can also be found in Shi and Lohmann (2016).

### 2.1.1 Idealised Earth system without tilted Earth rotation axis

In the first experiment, we create an idealised Earth system without seasonal migration of Earth's subsolar point (obliquity=0), while introducing a relatively elliptical Earth's orbit (eccentricity=0.058). In this experiment, perihelion takes place in June and aphelion takes place in December. We run the simulation for 1000 years, with the last 100 years result used for analysis. This experiment is designed to examine how changing the Earth-Sun distance, or perihelion, affects tropical precipitation over land.

### 2.1.2 Simulations reconstructing a precessional cycle

In the second experiment, we perform a set of simulations to reconstruct the climate change within a precessional cycle. In these simulations, the obliquity is set to be 24.5°. Twenty-four sensitivity simulations were ran using different precessional phases, i.e., 0°, 15°, 30°, . . . 345°. To allow the low-latitude climate to reach a quasi-equilibrium state to insolation forcing, we integrated the model for 300 years, and the last 100 years mean climate were used to represent the climate at the corresponding precession phase.

All these 24 simulations are initialized from a pre-industrial simulation which had participated in the PMIP4 project (Kageyama et al., 2018). For all simulations, the greenhouse gas concentrations in the atmosphere, land-sea distribution and ice sheet configuration are fixed as the pre-industrial condition.

Natural climate variability is ubiquitous that sometimes conceals external forcing. To highlight the external precessional forcing in climate change, in all simulations, the eccentricity is set to a relatively high value (0.058), which represents the highest eccentricity during the quaternary (Berger, 1978). Consequently, this configuration exposes the Earth to 26% more solar radiation during perihelion than during aphelion. Additionally, we examine the climatology mean of the last 100 years model results. This further eliminates the climate noise from natural variability.

The definition of seasonality, which is influenced by slow variations in the Earth's orbit, plays a key role in determining the calculated seasonal cycle of the climate. Application of the Gregorian calendar where the lengths of the months and seasons are fixed results in a drift in the occurrence date of different seasons. Especially, the applied high eccentricity (0.058) leads to a shift in the date of the autumn equinox by up to 27 days within our simulations. This may lead artificial biases when comparing monthly temperature and precipitation across different simulations with different precessional phases (Kutzbach and Gallimore, 1988; Joussaume and Braconnot, 1997). In contrast to the "fix-day" Gregorian calendar widely used today, the angular calendar calculates the lengths of the months and seasons according to a fixed angle along the Earth's orbit. When comparing simulation results for different orbital configurations, it is essential to use the angular calendar to ensure that the data

for comparison are from the same position along the Earth's orbit (Joussaume and Braconnot, 1997; Pollard and Reusch, 2002). To address this, we applied a calendar correction on our model results (temperature, precipitation, insolation) by changing the monthly mean data from the Gregorian calendar to an angular calendar. Detailed methodology can be found in Shi et al. (2022).

## 2.2 Speleothem proxies

With precise chronologies and widespread distribution, speleothem records are widely used to reconstruct the past hydrological cycle (Kaushal et al., 2024). To verify our hypothesis, we synthesize the speleothem $\delta^{18}O$ records from China, Malaysia and Brazil. The record from China is composited by Cheng et al. (2016) based on several samples from Sanbao (31°40' N, 110° 26' E), Hulu (32° 30' N, 119° 10' E), and Dongge (25° 17' N, 108° 5' E) caves (Wang et al., 2001; Dykoski et al., 2005; Kelly et al., 2006; Wang et al., 2008; Cheng et al., 2009). The Malaysia records were collected in different caves in Gunung Mulu National Park, Borneo, Malaysia ( 4°N, 115°E) (Carolin et al., 2013, 2016). The Brazil record is collected from the Botuvera Cave (27° 13' 24" S, 49° 09' 20" W) (Cruz Jr et al., 2005). These $\delta^{18}O$ records are believed to represent the changes in local precipitation amounts, as indicated by the original publications (Cruz Jr et al., 2005; Carolin et al., 2013; Cheng et al., 2016).

## 2.3 Method

### 2.3.1 Calculation of solar radiation pattern

To understand how the solar radiation fluctuates throughout a precessional cycle, we compute seasonal solar radiation patterns using MATLAB with aids from two tools, namely, "Orbital, the Box" (Lougheed, 2021) and the Earth Orbit Model v2.1 (Kostadinov and Gilb, 2014). These tools were developed based on the theoretical framework presented in (Berger, 1978; Laskar, 1990; Laskar et al., 1993).

### 2.3.2 Tracking the thermal equator

The ITCZ precipitation takes place over the Earth's warmest region, i.e., the thermal equator. To understand the movement of tropical precipitation, we track the thermal equator based on moist static energy (analogous to equivalent potential temperature) using the following equation:

$$S = C_p \cdot T + g \cdot z + L_v \cdot q \tag{1}$$

where $S$ is the moist static energy, $C_p$ is the specific heat at constant pressure, $T$ is the absolute air temperature, $g$ is the gravitational constant, $z$ is the geopotential height above sea level, $L_v$ is the latent heat of vaporization, and $q$ is water vapor specific humidity (Neelin and Held, 1987; Wallace and Hobbs, 2006). The region with the highest moist static energy is the thermal equator, where deep convection occurs (Barry and Chorley, 2009).

### 2.3.3 Empirical orthogonal functions analysis

The Empirical Orthogonal Functions (EOF) analysis (Hannachi et al., 2007) is used to identify the spatial and temporal characteristics of terrestrial precipitation at low latitudes. EOF analysis is widely applied in Earth science. It is generally used to simplify a spatial-temporal data set by converting it into spatial patterns of variability and temporal evolution of these patterns. For the idealised Earth system experiment without tilted Earth rotation axis, we applied EOF analysis on the climatology monthly convective precipitation over land between $40^o$S-$40^o$N (Fig. 4). For the 24 simulations recovering a precessional cycle, we applied EOF analysis on the individual monthly convective precipitation over land. For example, December precipitation in the 24 simulations is selected and applied a EOF analysis to generate Fig. 5a. Area weighting is not applied in the calculation.

## 3 Precession shifts the season and latitude of perihelion

To understand the mechanism of how precession governs the low-latitude hydrological cycle, we first look at how precession affects the solar radiation received by the Earth.

The Earth's orbit around the Sun is not a perfect circle but an ellipse. When the Earth's distance from the Sun is at its shortest, i.e., perihelion, it receives the strongest solar radiation. Currently, perihelion happens in boreal winter, when the Earth's subsolar point closes to the Tropic of Capricorn (Fig. S2d). By changing the orientation of the Earth's rotation axis, precession gradually delays the calendar timing of perihelion by around 25.1 minutes per year. This is equivalent to the Earth's subsolar point at perihelion migrating by about 500 meters per year. About 11 kiloyears ago, perihelion occurred in boreal summer, when the Earth's subsolar point was closest to the Tropic of Cancer (Fig. S2b). Therefore, precession not only shifts the calendar timing of perihelion but also the "latitude of perihelion". Here, we introduce the latitude of perihelion, which is the latitude of Earth's subsolar point during perihelion (Fig. S2). This latitudinal zone represents the region with the most intense incoming solar radiation. Logically, it also corresponds to the strongest thermal equator if the solar heating effect is instantaneous.

To detail when and where perihelion occurs, Fig. 3 shows the calendar timing and latitude of perihelion at different precessional phases (longitude of perihelion). Similar to the movement of the Earth's subsolar point over the course of a year, the latitude of perihelion migrates between the Tropics of Cancer and Capricorn over the course of a precessional cycle. Supplementary Movie 1 further clarifies this. Whenever (the season) and wherever (the latitude) perihelion occurs, the solar radiation in the corresponding season and the perihelion latitude reaches its maximum in a precessional cycle.

## 4 Perihelion promotes tropical convective precipitation over land

To investigate how perihelion impacts low-latitude climate change, we conducted an idealised Earth system experiment using AWI-ESM. In this idealised Earth system, the Earth's rotation axis is not tilted (zero obliquity). The Earth's orbit is set to be relatively elliptical (see Section 2.1.1). As a result, the thermal equator and the associated tropical rain belt are constrained in

the vicinity of the geographic equator. However, due to variations in the distance between the Earth and the Sun, temperature and precipitation exhibit a seasonality (Fig. 4).

At perihelion, the incoming solar radiation maximises. Due to the different thermal inertia between the land and the ocean, the atmospheric heating over land is stronger than that over the ocean. This leads to a faster increase in moist static energy over land than over the ocean (Fig. S3). As tropical deep convection tends to occur towards the warmest regions with the highest moist static energy (Neelin and Held, 1987; Battisti et al., 2014; Schneider et al., 2014; Geen et al., 2020), we observed maximum convective precipitation over land 1-2 months after the perihelion (Fig. 4). In brief, perihelion promotes tropical precipitation over land. With this understanding, we switch to a more complex Earth system with a tilted Earth's rotation axis and shifting perihelion.

## 5   Seasonal terrestrial precipitation peaks whenever and wherever perihelion occurs

The above experiment depicts a very idealized Earth. To simulate the climate in a more realistic world, we conducted another set of numerical experiment using AWI-ESM. This time, the Earth's rotation axis is tilted (obliquity=24.5°). Therefore, the Earth's subsolar point marches between the Tropics of Cancer and Capricorn, driving a seasonal migration of ITCZ and terrestrial precipitation (Fig. 1). We manipulated precessional phase to reconstruct the climate changes over the course of a precessional cycle (see Section 2.1.2). Specifically, 24 time-slice simulations were performed with the precessional phase (namely the longitude of the perihelion) varying from 0° to 345° with an interval of 15° (Fig. 2). Due to different precessional phases, when (the calendar timing) and where (the latitude) the perihelion takes place vary among these experiments (Fig. 3).

Because low-latitude land precipitation is concentrated at different latitudes in different months (Fig. 1), we investigate how precipitation evolves in each month over a precession cycle. EOF analysis was applied to explore the spatial and temporal characteristics of terrestrial precipitation oscillation. This allows us to identify where (the geographical location) and when (the precessional phase) the terrestrial precipitation peaks.

In February, the ITCZ and predominant terrestrial precipitation locates at its southernmost latitudes (Fig. 1 and 5f, contours). EOF analysis on the February precipitation reveals that the strongest terrestrial precipitation is around precessional phase of -75° (Fig. 5f, blue line). This corresponds to the perihelion occurring 1.5 month earlier (in early January, as shown in Fig. 5f with a colored circle) at its southernmost latitudes.

The ITCZ and associated rain belt locates around the equator in April (Fig. 1 and 5e, contours). We found the strongest April precipitation around the precessional phase of 15° (Fig. 5e, blue line) when perihelion occurs in March, with the latitude of perihelion being around the equator (Fig. 5e, colored circle).

Following the seasonal migration of Earth's subsolar point, the rain belt moves northward and reaches its northernmost position in August (Fig 1 and 5c, contours). Within a precession cycle, August terrestrial precipitation peaks near precessional phase of 120° (Fig. 5c, blue line), corresponding to a late July perihelion. This means that the perihelion latitude is almost at its northernmost position (Fig. 5c, colored circle).

From August to February, the terrestrial precipitation migrates southward, completing an annual cycle (Fig. 1). Whenever (the season) and wherever (the latitude) perihelion occurs, the related seasonal and latitudinal terrestrial precipitation reaches its maximum. This leads asynchronous terrestrial precipitation optimum at different seasons and latitudes (Fig. 5, blue arrow in the left panels).

There is good co-variation between the seasonal insolation intensity (Fig. 5, black lines), land surface temperature (Fig. 5, red lines) and terrestrial precipitation (Fig. 5, blue lines). We propose that perihelion maximizes the incoming solar radiation and drives the greatest land-sea heating contrast. This thermodynamically moves the thermal equator and the corresponding tropical convection from ocean to land (Fig. S4) (Battisti et al., 2014), and forces the strongest seasonal terrestrial precipitation within a precessional cycle. The simulated increased/decreased precipitation over land/ocean after the perihelion is supported by geologic records, which show decreased/increase $\delta^{18}O$ over land/ocean (Cheng et al., 2016; Huang et al., 2020; Jian et al., 2022). Due to the Earth's thermal inertia, the thermal equator and seasonal terrestrial precipitation are not synchronized with the insolation in the same month, but the insolation about 1-2 months earlier. In agreement, such a delay has been also revealed by many other model simulations (Kutzbach et al., 2008; Erb et al., 2015; Donohoe et al., 2020; Liu et al., 2022).

In a precessional cycle, the perihelion shifts toward different seasons and perihelion latitudes. This results in the strongest solar radiation and terrestrial precipitation in the corresponding season and latitudes (Fig. 6). Theoretically, for the tropical latitudes between the Tropics of Cancer and Capricorn, there are two passages of perihelion latitude within a precession cycle. For instance, at the equator, the first perihelion passage occurs at precessional phase of 0° around the spring equinox (Fig. 3). The second perihelion passage occurs at precessional phase of 180° around the autumnal equinox (Fig. 3). Consequently, at the equator, the temporal interval between these two perihelion passages spans half of a precessional cycle, resulting in cyclic variations in precipitation with a period of half of a precession cycle (Fig. 6c).

For the area outside the equatorial zone, the time interval between the two successive perihelion passages varies across different latitudes (Fig. 3 and Supplementary Movie 1). With increasing latitude, these two perihelion passages take place closer to each other, and merge into a single insolation and precipitation maximum at latitudes higher than the Tropic of Cancer (Fig. 3). This phenomenon is evident in our analysis of monthly precipitation patterns derived from zonal mean precipitation in different latitudinal belts (Fig. 6).

It is worth to mention that we have conducted an analysis of the precipitation using a zonal mean framework. Nevertheless, region-specific variations in the precipitation signal are evident in Fig. 4. These regional characteristics are obviously affected by the land-sea distribution and topography effect.

## 6    Evidence from geologic records

Besides model simulations, a realistic precession cycle can also be examined in the sparse-distributed geologic archives. There is a relatively abundant collection of proxies for the most recent precession cycle, spanning the past 23 ka. However, these records may not be ideal for deriving precession-associated climate changes, due to the limitation of the relatively low eccentricity (Berger, 1978, 1988), which diminishes the influence of precession on driving climate variations (Fig. 7) (Braconnot

and Marti, 2003; Bosmans et al., 2018; Chiang et al., 2022; Beaufort and Sarr, 2023; Wu et al., 2023a). Moreover, the scenario becomes more complex due to a variety of factors, such as the rapid disintegration of ice sheets, rising sea level, increasing greenhouse gas concentrations, and abrupt climate changes (Griffiths et al., 2009; Weber and Tuenter, 2011; Wang et al., 2014a; Chen et al., 2015; Chiang and Friedman, 2012; Clemens et al., 2021). Given the amplification of the precessional signal during periods of high eccentricity (Chiang and Broccoli, 2023) (Fig. 7), our investigation focused on the interval spanning 66-133 ka, thus encompassing three complete precessional cycles.

Many proxies' chronology is astronomically tuned to the Northern Hemisphere summer insolation, making them inappropriate to address the asynchronous signals at different latitudes. Here, we utilize the absolute-dated speleothem $\delta^{18}O$ records which were widely used as proxies of precipitation amount, despite the facts that they are somewhat also affected by other factors (Dykoski et al., 2005; Lachniet, 2009; Cai et al., 2010; Fairchild and Baker, 2012; Parker et al., 2021). We synthesize proxies from China (Cheng et al., 2016), Malaysia (Carolin et al., 2013, 2016) and Brazil (Cruz Jr et al., 2005) to test our hypothesis (Fig. 2). These records are selected because they are representative of three typical regions (i.e., northern limit, equator and southern limit of low-latitude) and have continuous long coverage and high-resolution Uranium-Thorium chronologies. Moreover, these records also well document the precessional dominated precipitation variations.

Tropical speleothem $\delta^{18}O$ records typically reflect an amount-(or infiltration) weighted annual mean precipitation $\delta^{18}O$, and are therefore typically biased towards the rainy season (Kwiecien et al., 2022; Liu et al., 2022). Modern observations indicate that the rainy season near the southeast China is boreal summer, whereas the dominant rainy season in Brazil is boreal winter. In the vicinity of Malaysia, there are two prevailing rainy seasons, namely the boreal spring and autumn (Fig. S1). Accordingly, we expect that the $\delta^{18}O$ signals in the selected regions reflect precipitation changes in their corresponding rainy seasons.

As depicted in Fig. 2b, the perihelion latitude reaches its northernmost position at 90° precessional phase (Fig. 2b, dashed line, and Fig. 3 and S2b), corresponding to maximum insolation in boreal summer and maximum in Eastern Asian summer precipitation. Around the precessional phase of 180°, the latitude of perihelion is close to the equator, corresponding to maximum insolation in autumnal equinox (Fig. 3 and S2c). The maximum boreal autumn insolation drives the maximum equatorial precipitation, as evidenced by the speleothem record from Malaysia around 121.7ka and 99.9 ka (Fig. 2c). As perihelion migrates to its southernmost position, i.e., at precessional phase 270° (Fig. 3 and S2d), the precipitation optimum is identified in Brazil, in agreement with the boreal winter insolation maxima (Fig. 2d). At a precessional phase of 0°, perihelion occurs around the equator in spring equinox (Fig. 3 and S2a). This leads to maximal insolation in March (Fig. 2c, grey line). We find maximum precipitation signals in the Malaysia speleothem record (Fig. 2c, red line) around 110.9 and 88.5 ka, corresponding to two maxima in March insolation. The perihelion moves to its northernmost position at precessional phase of 90°, thus completing a full precessional cycle. For the East Asian and the South American summer monsoons, they reach their minima during the seasonal insolation minima. In contrast, at the equator, the minimum precipitation occurs not during seasonal insolation minima but during the time when insolation is relatively weak in both rainy seasons, i.e., at the precession phases of 90° and 270° (Fig. 6c), approximately 116.2, 105.5, and 82.8 ka (Fig. 2c).

Regionally, there are not perfect agreements between the seasonal insolation variations and speleothem records. For example, the Chinese proxies peak at 90° longitude of perihelion around 127.3 ka. It stays high until after 121.7 ka. This long-term

precipitation optimum was likely maintained by relatively high $CO_2$ during the last interglacial period. Moreover, millennial-scale abrupt climate changes originating from the North Atlantic also play a role in shaping the precipitation across the low-latitudes (Wang et al., 2008; Chiang and Friedman, 2012). This is evident in the abrupt jumps in $\delta^{18}O$ signals from all different latitudes (Fig. 2). In addition, we noticed that the manifestation of October insolation in the half precessional signals in the Malaysia record is relatively stronger than that of the insolation in March. This is likely attributed to a stronger boreal autumn rain than the boreal spring rain in Malaysia (Fig. S1).

The results based on the speleothem records suggest that wherever the perihelion occurs, the local terrestrial precipitation peaks. This exhibits an asynchronous precipitation optimum following the meridional migration of perihelion (Fig. 1, dashed line). Near the equator, half-precessional signals are identified, in agreement with our model simulations (Fig. 6) and many other records from the tropical area (Trauth and Strecker, 1996; Trauth et al., 2003; Verschuren et al., 2009; Jian et al., 2020).

It is interesting to note that shifting perihelion causes changes in the rainy seasons. An earlier study found that the shift of maximum summer insolation has a strong impact on the Asian monsoon (Braconnot and Marti, 2003). Boreal spring perihelion leads early northward migration of the Northern Hemisphere westerly jet (Wu et al., 2023a). In our simulations, we identify that the dominant rainy season shifted from June to September near the 10°N latitudinal belt. At the equator, the dominant rainy season switches from April to October following the shift in perihelion (Fig. 6). This could complicate the comparison between proxies and models, as variations in the rainy seasons are difficult to distinguish by proxies (Kwiecien et al., 2022).

## 7 Discussions and Conclusion

In this study, we investigate the dynamics of asynchronous evolution of low-latitude precipitation under precession forcing, a long-standing conundrum in paleoclimate research. We hypothesize that the precessional-scale low-latitude hydrological cycle is paced by shifting perihelion rather than the hemispheric summer insolation. Using two sets of idealized climate model simulations, we showed that whenever and wherever perihelion occurs, the tropical terrestrial precipitation peaks in the corresponding perihelion season and latitudinal band. Under this new framework, the low-latitude precipitation naturally follows distinct rhythms. Speleothem proxies from three typical latitudes were included to test our hypothesis, which shows that the regional precipitation optimum matches the meridional migration of perihelion latitude.

Previous studies have used transient simulations to investigate the dynamics of orbital-scale climate changes. These simulations introduced transient forcing attributed to changing Earth's orbit (Kutzbach et al., 2008; He, 2011; Singarayer et al., 2017; Liu et al., 2022), $CO_2$, land-sea mask, and ice-sheet configurations (Liu et al., 2009). Despite being more realistic, the simulations with various forcing factors have added difficulty to disentangling the dynamics of precession's impact on low-latitude climate changes (Chen et al., 2015; Cheng et al., 2021; Griffiths et al., 2009; Ran and Feng, 2013). Time-slice simulations with precession extremes were also performed (Kutzbach, 1981; Braconnot et al., 2008; Battisti et al., 2014; Erb et al., 2015; Bosmans et al., 2018; Jalihal et al., 2019) to assess the precession's role in climate change. However, they are unable to capture the diverse evolution patterns of low-latitude precipitation, as revealed by proxies (An et al., 2000; Wang et al., 2004; Cruz Jr et al., 2005; Cheng et al., 2013; Carolin et al., 2013, 2016; Zhou et al., 2022). In this study, we performed simulations purely

forced by a varying precessional phase. The results reveal that the low-latitude terrestrial precipitation maxima follow a shifting perihelion, with a time lag of 1-2 months.

Traditionally, low-latitude precipitation was regarded as a manifestation of global monsoon, which is usually defined as occurring in hemispheric summer (Wang and Ding, 2008; Wang et al., 2014a, 2017; Geen et al., 2020). Therefore, the inter-hemispheric summer insolation difference was considered as the main driver of precipitation changes at low latitudes (Wang et al., 2014a; Schneider et al., 2014). Under this framework, the non-summer precipitation has received less attention. In reality, the ITCZ-related precipitation occurs in different seasons (Fig. 4), not necessarily during the hemispheric summer. Therefore, a comprehensive hypothesis should explain precipitation changes not only in the hemispheric summer but also in other seasons. We find that shifting perihelion likely plays an important role in the fluctuations of seasonal and latitudinal tropical precipitation.

Several studies have shown that the distance effect, or perihelion and aphelion, has an impact on low-latitude seasonality in addition to the march of Earth's subsolar point (Braconnot et al., 2008; Chiang et al., 2022; Beaufort and Sarr, 2023; Wu et al., 2023a; Chiang and Broccoli, 2023; Hunt et al., 2023). Increased solar radiation can thermodynamically shift the tropical convergence zone from ocean to land (Battisti et al., 2014), thus the terrestrial precipitation is enhanced at perihelion. Perihelion occurs in different seasons and latitudes, driving enhancement of terrestrial precipitation in the corresponding seasons and latitudes. From this point of view, insolation in individual seasons is equally important in determining the evolution of low-latitude precipitation.

Astronomical tuning is widely used to establish the chronology of paleo proxies. By doing this, the phasing of proxies is artificially synchronized. However, our results indicate that the astronomically driven climate changes can naturally follow diverse rhythms. This questions the reliability of the astronomical tuning strategy. For example, absolutely dated proxies indicate an asynchronous onset and termination of Greening Sahara at different latitudes (Kuper and Kropelin, 2006; Hély et al., 2014; Shanahan et al., 2015). Synchronizing the ages of proxies from different regions may lead to biases of a few millennia and introduce difficulties in understanding their dynamics. Based on our model simulations (Fig. 6), we suggest that astronomical tuning should target the insolation 1-2 months before the local rainy season, at least for precipitation-related proxies in low-latitudes.

We evaluated our hypothesis using limited speleothem $\delta^{18}O$ records as proxies of the precipitation amount. However, speleothem $\delta^{18}O$ signals can be influenced not only by changes in precipitation amount but also by a variety of other factors (Baker et al., 2019) , such as the moisture source (Maher and Thompson, 2012), the transport pathway (Griffiths et al., 2009; Wurtzel et al., 2018), the degree of upstream precipitation (Cheng et al., 2013; Shi et al., 2025), atmospheric circulation (Breitenbach et al., 2010; Sinha et al., 2015), cave microclimate (Vieten et al., 2016; Sekhon et al., 2021; Treble et al., 2022; Patterson et al., 2024), or a combination of these processes (Dykoski et al., 2005; Lachniet, 2009; Fairchild and Baker, 2012; Baker et al., 2019; Parker et al., 2021). This causes additional uncertainties. Therefore, validating using a wide range of additional proxies is necessary and welcomed.

The present study focuses solely on how precession affects the low-latitude hydrological cycle. Besides precession, many other factors also contribute to shaping the ICTZ precipitation. For example, high obliquity increases the hemispheric summer

insolation, thereby contributing to an increase in hemispheric monsoon precipitation (Erb et al., 2015; Bischoff et al., 2017; Bosmans et al., 2018). The presence of high-latitude ice sheets introduces hemispheric cooling, moving the ITCZ away from the cold hemisphere (Chiang and Bitz, 2005; Weber and Tuenter, 2011; Chen et al., 2015; Clemens et al., 2021; Wu et al., 2023b). Similarly, abrupt North Atlantic cooling events associated with the collapse of the Atlantic Meridional Overturning Circulation drive the southward shift of the ITCZ (Wang et al., 2008; Chiang and Friedman, 2012; Singarayer et al., 2017). Sea level fluctuations alter the land-sea distribution, affecting the supply of moisture, and therefore regional precipitation (Griffiths et al., 2009). The combination of these factors results in a complex evolution of precipitation changes (Bischoff et al., 2017; Lyu et al., 2021; Yuan et al., 2023), which may also give rise to asynchronous precipitation signals throughout low-latitude regions.

By detailing the position and seasonal timing of perihelion (Fig. 3), the precession-induced terrestrial precipitation maxima can be easily predicted at a given latitude and season. Currently, perihelion takes place around the Tropic of Capricorn; the southern hemisphere summer monsoon is relatively strong. Over the next few thousand years, following the movement of perihelion towards the equator, the equatorial terrestrial precipitation around the spring equinox will be enhanced. Compared to the classical theory, which highlights the role of summer insolation in driving a synchronous ITCZ migration, we hypothesize an asynchronous nature of low-latitude precipitation optimums following the shifting perihelion. This offers a more plausible explanation for the observed asynchronous pattern of low-latitude precipitation's response to precessional forcing.

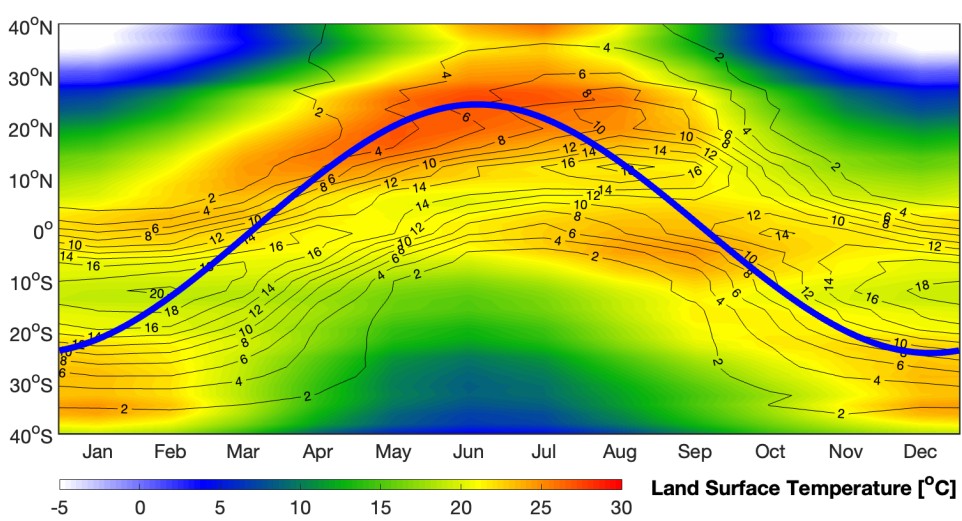

**Figure 1.** Seasonal migration of the Earth's subsolar point (thick blue line), thermal equator over land (shading color) and precipitation over land (black contours). Shading colors represent zonal mean land surface temperature. Contours show the zonal mean precipitation on land (unit: mm/day). The seasonal movement of the Earth's subsolar point fundamentally determines the position of the Earth's thermal equator, where deep convection and low-latitude precipitation takes place. Due to the thermal inertia of Earth, the thermal equator and land precipitation do not precisely coincide with the timing of maximal solar radiation, i.e., the Earth's subsolar point, but rather experience a delay of approximately 1-2 months. Results are based on the ensemble mean of the 24 simulations with different precessional phases (see Methodology).

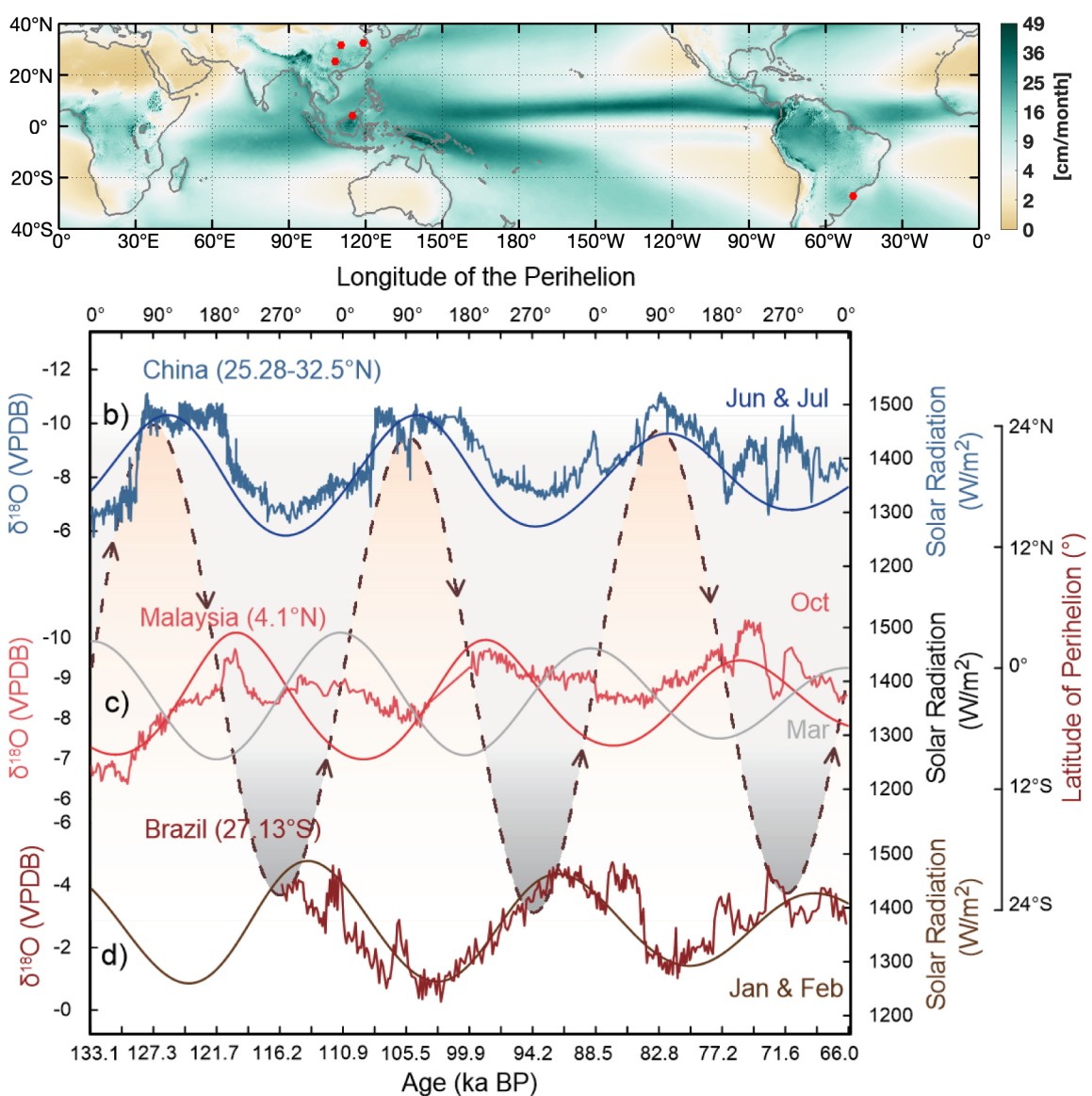

**Figure 2.** (a) Climatology map of annual mean precipitation over low-latitudes. Stalagmite oxygen isotope evolution in (b) China (Cheng et al., 2016) (c) Malaysia (4.1°N, 114.9°E) (Carolin et al., 2013, 2016) and (d) Brazil (27°13' 24" S, 49°09' 20" W) (Cruz Jr et al., 2005). The locations of these records are shown in Fig. 2a. The intensity of solar radiation in the rainy seasons are plotted as well: for China, it is the averaged solar radiation in June and July; for Malaysia, it is the solar radiation in October and March; for Brazil, it is the mean solar radiation in January and February. The dashed line illustrates the meridional migration of perihelion latitude (see detailed definition in Section 3). To better illustrate the terrestrial precipitation optimums at different precessional phases, we translate the age into precessional phase, which is shown in the upper axis. When the perihelion passes certain latitude, the seasonal solar radiation and terrestrial precipitation reach their maxima.

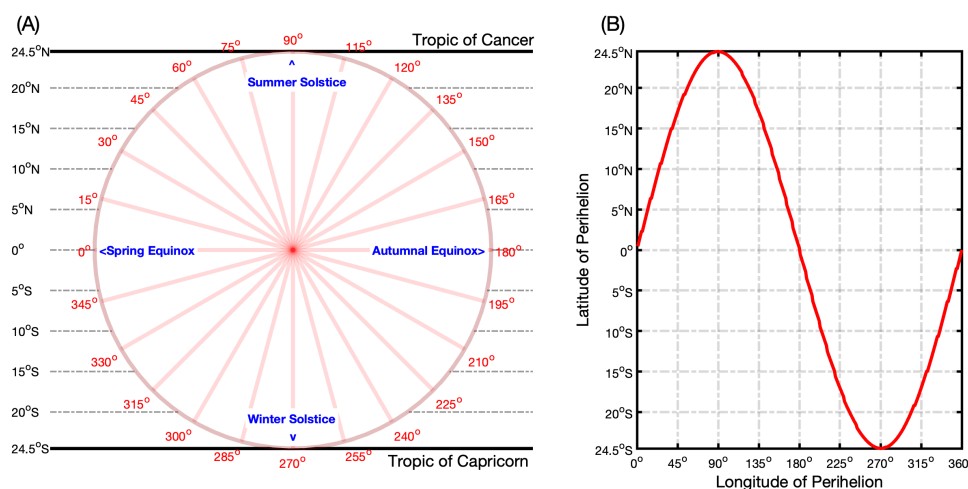

**Figure 3.** The latitudinal movement of the perihelion and Earth's subsolar point within a precessional cycle and an annual cycle, respectively. The labels outside the circle give the precessional phase, namely the longitude of the perihelion. The labels inside the circle locate the position of season. Within a year, the Earth's subsolar point march between the Tropic of Cancer and the Tropic of Capricorn. Within a precessional cycle, the latitude of perihelion also migrates between the Tropic of Cancer and the Tropic of Capricorn. The grey dashed lines give the latitudinal coordinate. When the perihelion occurs, the insolation at the perihelion latitude reaches the maximum within a precessional cycle. To better understand this figure, please also check Supplementary Movie 1 (https://zenodo.org/doi/10.5281/zenodo.11395458), which visually conveys an explanatory description of the seasonal solar radiation variations within a precessional cycle.

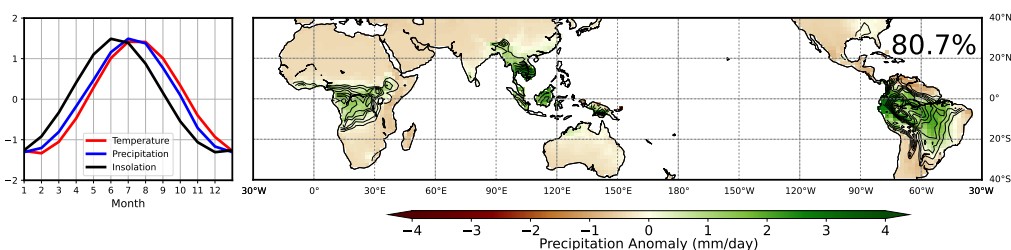

**Figure 4.** Temporal and spatial evolutions of terrestrial precipitation in the idealised Earth system experiment, with zero obliquity and high eccentricity (0.058). The shading color in the right panel illustrates the spatial pattern of the leading mode (contains 80.7% of terrestrial precipitation covariance) of Empirical Orthogonal Function (EOF) analysis on the monthly convective precipitation over land. The contours illustrate the climatology precipitation. The blue lines in the left panels show the corresponding principal component, illustrating the temporal evolution of the terrestrial precipitation strength. The area-weighted land-surface temperature (red line) and incoming solar radiation (black line) evolution are plotted as well. In this experiment, perihelion occurs in June, corresponding to maximum incoming solar radiation in June and maximum precipitation over land 1-2 month later (July and August).

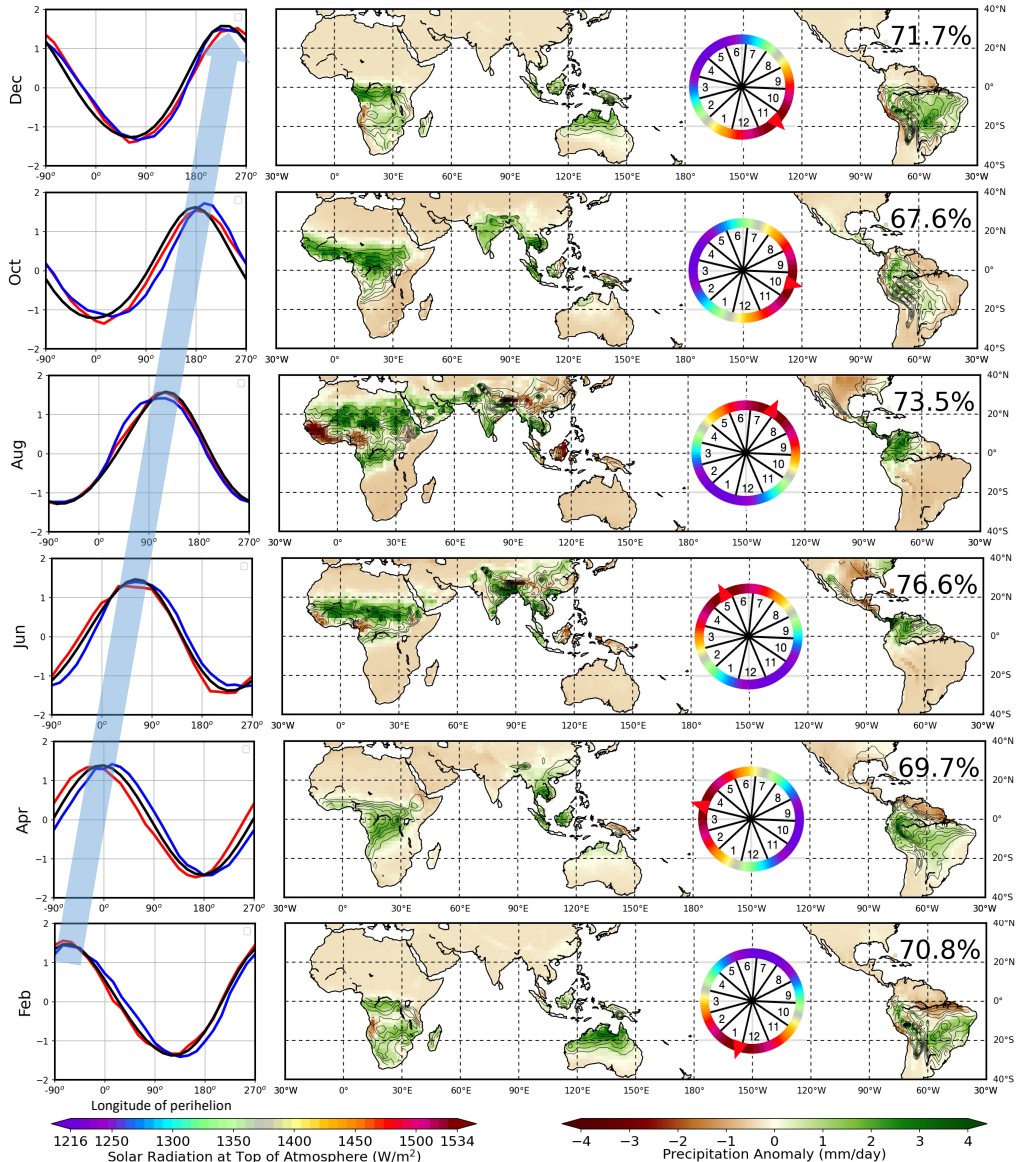

**Figure 5.** Temporal and spatial evolutions of terrestrial precipitation in different seasons. The shading color in the right panel illustrates the spatial pattern of the leading mode of Empirical Orthogonal Function (EOF) analysis on the monthly terrestrial precipitation simulated by AWI-ESM recovering a precessional cycle. The numbers in the subpanels' upper-right corner displays the contribution of the first principal component of EOF to the overall variance. The contours illustrate the climatological precipitation in the corresponding month. The blue lines in the left panels show the corresponding principal component of EOF analysis, illustrating the temporal evolution of the terrestrial precipitation strength. The area-weighted land (between 40$^o$S-40$^o$N) surface temperature (red line) and global mean incoming solar radiation (black line) evolution are plotted as well. Note that, the plotted solar radiation is from one month before. For example, for the panel of February/August, the solar radiation in January/July is plotted. For easy comparison, all time-series are normalised. The colored circles illustrate the solar radiation distribution during the precessional phase when the corresponding seasonal terrestrial precipitation peaks. The numbers in this circle give the location of the individual month. The red arrows point the timing and meridional position of perihelion. The blue arrow in the left panels illustrates the asynchronous precipitation optimums across different latitudes and seasons.

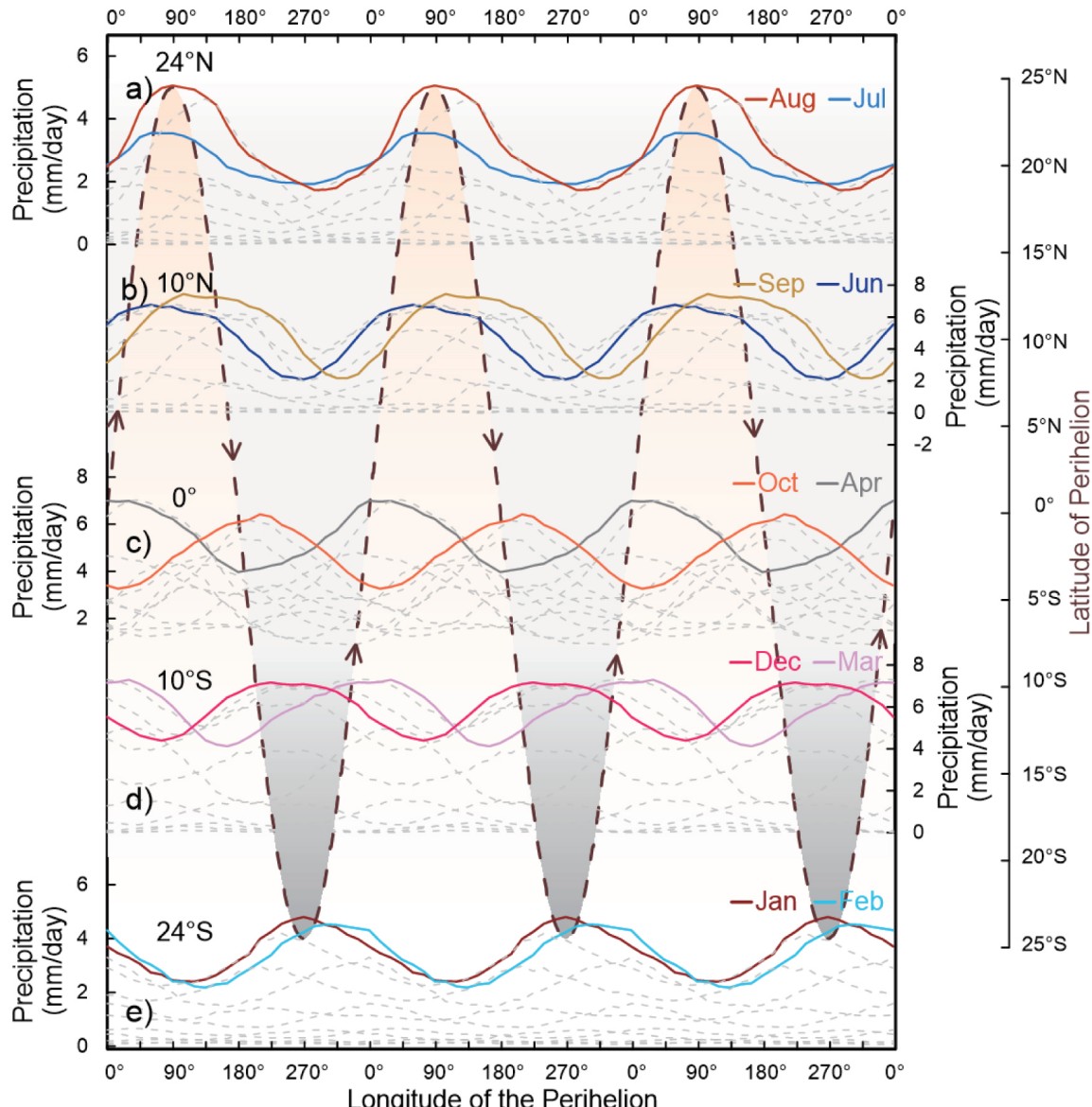

**Figure 6.** The simulated evolution of monthly precipitation at different latitudinal bands over three precessional cycles. The precipitation is calculated as zonal mean terrestrial precipitation over latitudinal bands covering a width of five degrees. For each latitudinal band, the rainy seasons are highlighted with colored lines, and precipitation in other months is shown as grey dashed lines. The dashed copper line illustrates the meridional movement of Earth's perihelion between the Tropic of Cancer and the Tropic of Capricorn, namely the latitude of perihelion. Whenever (the season) and wherever (the latitude) perihelion occurs, the local terrestrial precipitation in the corresponding month reaches its maximum within a precessional cycle.

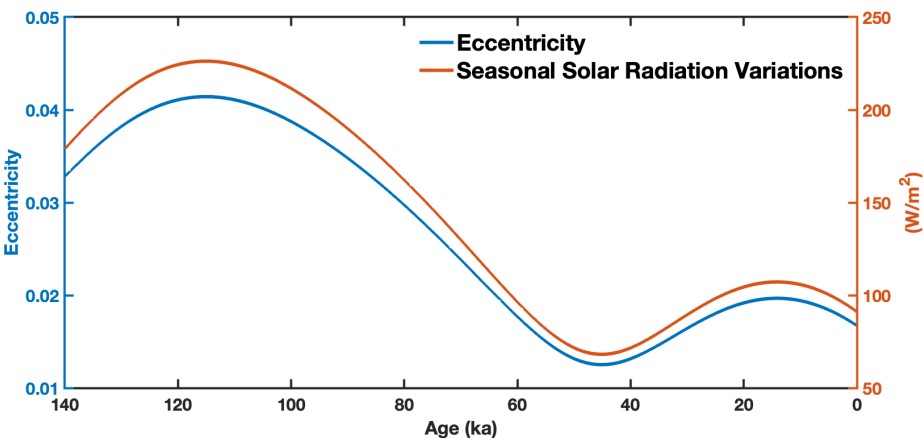

**Figure 7.** Eccentricity and magnitude of seasonal variations in solar radiation intensity over the past 140 ka. The eccentricity is calculated according to Berger (1978), and the amplitude of seasonal incoming solar radiation intensity is computed as the difference between the solar radiation at perihelion and aphelion.

*Data availability.* The model output used in this study can be accessed from https://zenodo.org/doi/10.5281/zenodo.13681175.

*Author contributions.* H. Yang conceived the idea and wrote the first draft. All authors participated in the manuscript's discussion and
revision.

*Competing interests.* The authors have declared that no competing interests exist.

*Acknowledgements.* We acknowledge Marie-France Loutre and the other two anonymous reviewers for providing valuable comments to improve the manuscript. This study was supported by Southern Marine Science and Engineering Guangdong Laboratory (Zhuhai), No. SML2023SP204, the Ocean Negative Carbon Emissions (ONCE) Program and the National Natural Science Foundation of China. The
AWI-ESM simulations were conducted on Deutsche Klimarechenzentrum (DKRZ) and AWI supercomputer (Ollie).

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
