# Peer review of "Precession-driven low-latitude hydrological cycle paced by shifting perihelion"

_EGUsphere, 2024_

## Author Response (AR1)

**Response Letter**

Dear Editor and Reviewers,

Thank you for your valuable time and efforts in reviewing our manuscript. According to your comments, we have made necessary changes. Here are the point-by-point answers to your comments.

Regards,

Hu Yang

**Reviewer #1**

This is a very technical comment on the paper but not at all a review on the scientific findings, and their validity.

The authors wrote "The latitude of the perihelion is introduced as the latitude of Sun's zenith point when perihelion occurs. This latitude also represents the latitude of maximum incoming solar radiation at the top of atmosphere."

As far as I know the zenith is defined for an observer on the Earth but not for the Sun. What the authors mean is probably something else. Is it the angular distance of the Sun (at the perihelion) from the zenith, i.e. co-latitude?

« the latitude of maximum incoming solar radiation » Do the authors mean on the day that the Sun reached the perihelion? In that case when perihelion occurs at summer/winter solstice the maximum incoming solar radiation is at the pole (north/south), not between the tropics.

A: As non-native English speakers, we apologize for any inaccurate wording. Following your suggestion, we have modified the sentence as follow:

*Line 10-11: "Here, the latitude of perihelion is introduced as the latitude of Earth's subsolar point during perihelion, which is the location where the most intense solar radiation is concentrated."*

*Line 155-159: "Therefore, precession not only shifts the calendar timing of perihelion but also the "latitude of perihelion". Here, we introduce the latitude of perihelion, which is the latitude of Earth's subsolar point during perihelion (Fig. S2). This latitudinal zone represents the region with the most intense incoming solar radiation. Logically, it also corresponds to the strongest thermal equator if the solar heating effect is instantaneous."*

"Currently, perihelion happens in boreal winter … About 11 kiloyears ago, perihelion occurred in boreal summer … » My understanding is that it means 180deg in 11kyr, which does not correspond to « around 20.4 minutes per year ».

A: Thank you for pointing out this mistake. We have compared the sidereal year with the tropical year. In fact, the comparison should be between anomalistic year and tropical year, as Reviewers #3 have also pointed out. The difference is 25.1 minutes. We have revised this detailed information in the revision.

The authors should clearly be more careful in their explanation. For example, they wrote 'when perihelion occurs'. As soon as the eccentricity is not zero, there is a perihelion. Therefore, it always 'occurs'. The authors probably meant something else but it is unclear what.

A: Perihelion does occur every year. But it occurs at different calendar time and latitude in every different year. In the revision, we have revised these sentences to

*"at the time of perihelion", 'during perihelion', 'at perihelion'*

**Reviewer #2**

The paper presents a hypothesis on how precession drives the low-latitude hydrological cycle, arguing that it is paced by shifting perihelion rather than hemispheric summer insolation. The content appears to be relevant to the field of paleoclimatology and orbital forcing of climate.

In general, the authors provide a comprehensive background on some classical theories and observations. The authors highlight limitations of classical theory in explaining asynchronous precipitation patterns observed in proxy records. Their hypothesis offers a plausible explanation that addresses these inconsistencies. The study's strengths lie in its multi-faceted approach, combining theoretical analysis, climate modeling, and proxy evidence. However, improvements could be made in the discussion of uncertainties, organization of results and discussion sections, and some aspects of writing clarity and citations.

A: Thank you for your valuable time in reviewing our manuscript. According to your suggestion, we have added several paragraphs to discuss the uncertainty in the speleothem proxies (line 315-321), as well as highlight the other possible mechanisms (line 322-331).

Also, following the suggestions from the first reviewer and you, we have reworded in the definition of latitude of perihelion and the Earth's subsolar point.

Regarding the organization of discussion. The original paper was submitted to a journal with limited space; therefore, we did not spend words to give a summary. In the revision, we have revised the discussion to give a brief conclusion of the paper and then discussed the uncertainties, and implications of the study. We hope the revised version satisfied you.

**Structure**

Overall, the paper presents a well-structured argument for the proposed hypothesis. The paper follows a logical structure which allows readers to easily follow the progression of ideas. The authors effectively use subheadings to guide the reader through different aspects of their study. One suggestion for improvement would be to more clearly delineate the transition between results and discussion sections. While the current structure works, a more explicit separation could help readers distinguish between the presentation of findings and their interpretation.

A: Following your suggestion, we have added one paragraph (Line 281-287) to summaries the findings of our study, and then discussion on the uncertainties and implications.

*Line 281-287: In this study, we investigate the dynamics of asynchronous evolution of low-latitude precipitation under precession forcing, a long-standing conundrum in paleoclimate research. We hypothesize that the precessional-scale low-latitude hydrological cycle is paced by shifting perihelion rather than the hemispheric summer insolation. Using two sets of idealized climate model simulations, we showed that whenever and wherever perihelion occurs, the tropical terrestrial precipitation peaks in the corresponding perihelion season and latitudinal band. Under this new framework, the low-latitude precipitation naturally follows distinct rhythms. Speleothem proxies from three typical latitudes were included to test our hypothesis, which shows that the regional precipitation optimum matches the meridional migration of perihelion latitude.*

**Methodology**

The authors effectively use a combination of theoretical analysis, climate model simulations, and comparison with geological records to support their claims. My impression is that the methodology employed in this study is overall appropriate and well-executed. In principle, the selected speleothem records from South America and Asia seem to be appropriate for the study, but it would be beneficial, if additional records could be included if available (e.g., when checking the SISAL database?).

A: Indeed, we have validated the results using limited proxies. We aware of this. Following your suggestion, we try to search more records, and find other proxies from the northeastern Brazil (10◦S), showing the local wet period is paced by boreal spring insolation (Wang et al., 2004). However, this proxy is not continuing long record. We have included it in the introduction.

We aware the limitation of study not using a large amount of data. There in the revised discussion, we have included discussion on the uncertainties of data and welcome for further study to test our hypothesis.

*Line 315-321: "We evaluated our hypothesis using limited speleothem δ18O records as proxies of the precipitation amount. However, speleothem δ18O signals can be influenced not only by changes in precipitation amount but also by a variety of other factors, such as the moisture source (Maher and Thompson, 2012), the transport pathway (Griffiths et al., 2009; Wurtzel et al., 2018), the degree of upstream precipitation (Cheng et al., 2013; Shi et al., 2025), atmospheric circulation (Breitenbach et al., 2010; Sinha et al., 2015), cave microclimate (Treble et al., 2022; Patterson et al., 2024), or a combination of these processes (Dykoski et al., 2005; Lachniet, 2009; Fairchild and Baker, 2012; Parker et al., 2021). This causes additional uncertainties. Therefore, validating using a wide range of additional proxies is necessary and welcomed."*

Concerning the model simulations, my expertise is limited, but my impression is that the experiments are well-designed, with an idealized Earth system experiment and a set of simulations reconstructing a full precessional cycle. However, one area that could be strengthened is the discussion of potential limitations or uncertainties in their model simulations and proxy interpretations. While the authors do mention some caveats, a more explicit treatment of uncertainties would enhance the robustness of their conclusions.

A: Following your suggestion, we have added more words on the limitation of our experiments and proxies in the discussion, as illustrated in

*Line 315-331: "We evaluated our hypothesis using limited speleothem δ18O records as proxies of the precipitation amount. However, speleothem δ18O signals can be influenced not only by changes in precipitation amount but also by a variety of other factors, such as the moisture source (Maher and Thompson, 2012), the transport pathway (Griffiths et al., 2009; Wurtzel et al., 2018), the degree of upstream precipitation (Cheng et al., 2013; Shi et al., 2025), atmospheric circulation (Breitenbach et al., 2010; Sinha et al., 2015), cave microclimate (Treble et al., 2022; Patterson et al., 2024), or a combination of these processes (Dykoski et al., 2005; Lachniet, 2009; Fairchild and Baker, 2012; Parker et al., 2021). This causes additional uncertainties. Therefore, validating using a wide range of additional proxies is necessary and welcomed.*

*The present study focuses solely on how precession affects the low-latitude hydrological cycle. Besides precession, many other factors also contribute to shaping the ICTZ precipitation. For example, high obliquity increases the hemispheric summer insolation, thereby contributing to an increase in hemispheric monsoon precipitation (Erb et al., 2015; Bischoff et al., 2017; Bosmans et al., 2018). The presence of high-latitude ice sheets introduces hemispheric cooling, moving the ITCZ away from the cold hemisphere (Chiang and Bitz, 2005; Weber and Tuenter, 2011; Chen et al., 2015; Clemens et al., 2021; Wu et al., 2023b). Similarly, abrupt North Atlantic cooling events associated with the collapse of the Atlantic Meridional Overturning Circulation drive the southward shift of the ITCZ (Wang et al., 2008; Chiang and Friedman, 2012). Sea level fluctuations alter the land-sea distribution, affecting the*

*supply of moisture, and therefore regional precipitation (Griffiths et al., 2009). The combination of*
*these factors results in a complex evolution of precipitation changes (Lyu et al., 2021; Yuan et al.,*
*2023), which may also give rise to asynchronous precipitation signals throughout low-latitude*
*regions."*

**Writing clarity**

While the writing is generally clear, there are a few instances where key concepts could be described with more precision or clarity (this is also related to the previous community comment…). For example, the statement about "the latitude of maximum incoming solar radiation at the top of atmosphere" oversimplifies the complex relationship between perihelion and insolation patterns, which depends on multiple factors including axial tilt. To improve clarity, the authors should carefully go through their MS again and reconsider such explanations, to provide more accurate and precise descriptions of key orbital mechanics concepts. This would strengthen the paper's foundation and help readers better understand the novel hypothesis presented.

A: As suggested, we revised the wording on the following two key messages. Please also checked the answers to the first reviewer.

Sun's zenith point. → Earth's subsolar point

maximum incoming solar radiation  → the most intense solar radiation

**Discussion**

In the discussion, the authors consider some alternative explanations and address potential weaknesses in their hypothesis. For example, they acknowledge factors like changing obliquity. However, I still consider their treatment of alternative explanations for asynchronous precipitation patterns as one area where the authors could improve in. While they mention various factors (e.g., ice sheets, vegetation feedback) that could disrupt summer insolation's control on the hydrological cycle, they don't engage deeply with these alternative explanations. A revised discussion should involve how the approach is able to combine thermodynamic and dynamic (atmospheric) processes, and how it compares to other works (e.g., Bischoff et al 2017, Singarayer et al. 2017, …).

A: As suggested, we have discussed the existing mechanism in an in-depth way. The recommended references were cited. See lines 323-333:

*"The present study focuses solely on how precession affects the low-latitude hydrological cycle.*
*Besides precession, many other factors also contribute to shaping the ICTZ precipitation. For example,*
*high obliquity increases the hemispheric summer insolation, thereby contributing to an increase in*
*hemispheric monsoon precipitation (Erb et al., 2015; Bischoff et al., 2017; Bosmans et al., 2018). The*

*presence of high-latitude ice sheets introduces hemispheric cooling, moving the ITCZ away from the cold hemisphere (Chiang and Bitz, 2005; Weber and Tuenter, 2011; Chen et al., 2015; Clemens et al., 2021; Wu et al., 2023b). Similarly, abrupt North Atlantic cooling events associated with the collapse of the Atlantic Meridional Overturning Circulation drive the southward shift of the ITCZ (Wang et al., 2008; Chiang and Friedman, 2012; Singarayer et al., 2017). Sea level fluctuations alter the land-sea distribution, affecting the supply of moisture, and therefore regional precipitation (Griffiths et al., 2009). The combination of these factors results in a complex evolution of precipitation changes (Bischoff et al., 2017; Lyu et al., 2021; Yuan et al., 2023), which may also give rise to asynchronous precipitation signals throughout low-latitude regions."*

Furthermore I miss in the discussion a statement, how representative the selected speleothem records are, given that previous works have shown differences between locations are observed on the precessional to orbital scale, (e.g., Windler et al 2021, Parker et al., 2021, Wu et al. 2023). In addition, care should be taken regarding fidelity of the $\delta^{18}$O proxy as a "precipitation proxy", and it should be clearly distinguished when the study is focusing on "precipitation amount" or "monsoon", which is not necessarily the same, and not always adequately represented in speleothem $\delta^{18}$O (e.g., Patterson et al., 2024, Wu et al., 2023).

A:We aware that we have used limited speleothem records, and there are also other factors controlling the $\delta^{18}$O signals in speleothem records. To addresses these uncertainties, we have included two paragraph to discuss the uncertainties as in lines 237-239 and lines 316-322.

*Line 237-239: "Here, we utilize the absolute-dated speleothem δ18O records which were widely used as proxies of precipitation amount, despite the facts that they are somewhat also affected by other factors (Dykoski et al., 2005; Lachniet, 2009; Cai et al., 2010; Fairchild and Baker, 2012; Parker et al., 2021)."*

*Line 316-322: "We evaluated our hypothesis using limited speleothem δ18O records as proxies of the precipitation amount. However, speleothem δ18O signals can be influenced not only by changes in precipitation amount but also by a variety of other factors, such as the moisture source (Maher and Thompson, 2012), the transport pathway (Griffiths et al., 2009; Wurtzel et al., 2018), the degree of upstream precipitation (Cheng et al., 2013; Shi et al., 2025), atmospheric circulation (Breitenbach et al., 2010; Sinha et al., 2015), cave microclimate (Treble et al., 2022; Patterson et al., 2024), or a combination of these processes (Dykoski et al., 2005; Lachniet, 2009; Fairchild and Baker, 2012; Parker et al., 2021). This causes additional uncertainties. Therefore, validating using a wide range of additional proxies is necessary and welcomed."*

I acknowledge that some of this discussion may be beyond the scope of this paper, but an overall more comprehensive discussion of the uncertainties and limitations of the approach and the hypothesis, as well as alternative models and explanations is very desirable.

A: As suggested, we have included more discussions on the uncertainties of our hypothesis and the speleothem records. We encourage future study to include more data and more complex model to test our hypothesis. Please checked the response to the previous comments.

**Minor comments**

L53 do you mean "presence"? Also these factors are mentioned here but not really discussed later how that fits to their hypothesis

A: Yes, we mean "presence of ice sheet". As suggested, we have provided more discussion on the other studies explain the asynchronous signals. Please checked line 323-333.

L57 affects

A: Corrected.

L63 strictly speaking, you are not constructing new geological records, but use already published data.

A: Yes, we have modified it to "… synthesis of geologic records from different latitudes", to indicate that we are using geologic records from others.

L63 to hypothesize

A: Corrected.

L98 ff It would be beneficial to reiterate for the individual records how the speleothem $\delta^{18}O$ is interpreted and how representative it is for local rainfall amount. Also have you checked the SISAL database (e.g., Kaushal et al., 2024) if there are more records available that could be included?

A: We are aware that we have included limited proxies. And these proxies may not fully represent precipitation amount. Following your suggestion, we have discussed the uncertainties in the revision, as in Line 116-123, Line 316-322. We have reference Kaushal et al., (2024) and encouraging future research to use more records to validate our results.

L119 Could this definition be described even more clearly? It is a bit confusing… But this is very crucial, possibly a sketch illustrating a few "screenshots" of different phases shown in the supplementary movie could make this clearer?

A: To be smooth, we have moved this paragraph to the results section. As suggested, we have included one more figure (Fig. S2) to illustrate the screenshots of different phases of precession and the corresponding perihelion latitude.

L132 see comment above.

A: We have taken your suggestion.

L134 To be honest, Fig. 2 is not very intriguing to someone who is not expert in orbital processes and associated notations. (compare also previous comments)

A: Yes, for the first view, it is difficult to understand. But it is likely the most accurate and comprehensive figure to include information of perihelion timing and latitude under different precession phases. We have provided both movie and another Figure (Fig. S2) to explain this.

L164ff How do the results explained in this paragraph compare to the so-called "classical theory"?

A: First, the "classical theory" only explains the precipitation changes in the hemispheric summer. The precipitation in other seasons is not included. Second, the classical theory proposed a hemispheric systematic response to summer insolation. While we proposed a natural asynchronous response of precipitation to precession. These differences are discussed in the last section of our revision. See lines 281-287 and lines 299-312.

*Line 281-287: "In this study, we investigate the dynamics of asynchronous evolution of low-latitude precipitation under precession forcing, a long-standing conundrum in paleoclimate research. We hypothesize that the precessional-scale low-latitude hydrological cycle is paced by shifting perihelion rather than the hemispheric summer insolation. Using two sets of idealized climate model simulations, we showed that whenever and wherever perihelion occurs, the tropical terrestrial precipitation peaks in the corresponding perihelion season and latitudinal band. Under this new framework, the low-latitude precipitation naturally follows distinct rhythms. Speleothem proxies from three typical latitudes were included to test our hypothesis, which shows that the regional precipitation optimum matches the meridional migration of perihelion latitude."*

*Line 299-312: Traditionally, low-latitude precipitation was regarded as a manifestation of global monsoon, which is usually defined as occurring in hemispheric summer (Wang and Ding, 2008; Wang et al., 2014a, 2017; Geen et al., 2020). Therefore, the interhemispheric summer insolation difference was considered as the main driver of precipitation changes at low latitudes (Wang et al., 2014a; Schneider et al., 2014). Under this framework, the non-summer precipitation has received less attention. In reality, the ITCZ-related precipitation occurs in different seasons (Fig. 4), not necessarily during the hemispheric summer. Therefore, a comprehensive hypothesis should explain precipitation changes not only in the hemispheric summer but also in other seasons. We find that shifting perihelion likely plays an important role in the fluctuations of seasonal and latitudinal tropical precipitation.*

*Several studies have shown that the distance effect, or perihelion and aphelion, has an impact on low-latitude seasonality in addition to the march of Earth's subsolar point (Braconnot et al., 2008; Chiang et al., 2022; Beaufort and Sarr, 2023; Wu et al., 2023a; Chiang and Broccoli, 2023; Hunt et al., 2023). Increased solar radiation can thermodynamically shift the tropical convergence zone from ocean to land (Battisti et al., 2014), thus the terrestrial precipitation is enhanced at perihelion. Perihelion occurs in different seasons and latitudes, driving enhancement of terrestrial precipitation in the corresponding seasons and latitudes. From this point of view, insolation in individual seasons is equally important in determining the evolution of low-latitude precipitation."*

L213 Just because the records document the precession-dominated variations, it doesn't demonstrate the selected records are truly representative for their latitudes.

A: Yes, you are right. The speleothem proxies are complex. Since the main scope of our study is to understand how precession affect the hydrological cycle. Therefore, we selected the proxies with dominant precessional signals. However, this does not mean that low-latitude precipitation must follow precession variation. As the real world is much more complex than precession. We have discussed the limitation of selected proxies and also the complex evolution of precipitation impacted by ice sheets, CO2, sea level changes, and abrupt climate changes. See lines 316-333.

L215 This is not true, that the growing season is always the rainy season. It depends on cave ventilation and vegetation activity, etc. In many cases growing season is winter. This is however generally independent on what the $\delta^{18}O$ in the speleothem represents, which is usually a (infiltration-weighted) annual mean value (e.g., Baker et al 2019) and can be hydrologically influenced (e.g., Treble et al 2022, Patterson et al 2024)). This interpretation is too generalized.

A: To weaken the statements, to included words of "typically", "usually", as below.

*"Speleothem records are typically biased toward their growing seasons, which are usually the rainy seasons (Kwiecien et al., 2022; Liu et al., 245 2022)."*

Also, in the discussed, we have discussed the uncertainty of speleothem records, and encouraging future studies to used wide range of other proxies to validate our results.

*Line 316-322: "We evaluated our hypothesis using limited speleothem δ18O records as proxies of the precipitation amount. However, speleothem δ18O signals can be influenced not only by changes in precipitation amount but also by a variety of other factors (Baker et al., 2019) , such as the moisture source (Maher and Thompson, 2012), the transport pathway (Griffiths et al., 2009; Wurtzel et al., 2018), the degree of upstream precipitation (Cheng et al., 2013; Shi et al., 2025), atmospheric circulation (Breitenbach et al., 2010; Sinha et al., 2015), cave microclimate (Treble et al., 2022; Patterson et al., 2024), or a combination of these processes (Dykoski et al., 2005; Lachniet, 2009; Fairchild and Baker, 2012; Baker et al., 2019; Parker et al., 2021). This causes additional*

*uncertainties. Therefore, validating using a wide range of additional proxies is necessary and welcomed."*

L235 This is based on only three records, possibly there are more available (see earlier comment)

A: As suggested, the uncertainties have been discussed, and we encourage future study to use wide range of other proxies to validate our results. Please check the answer to previous comments.

L260 with this conclusion, the authors should think about the overall wording if using the notation of "summer" (and similar) is precise and adequate at all instances...?

A: We have used hemispheric summer to infer both boreal summer and austral summer.

L276 This paragraph could be more expanded

A: As suggested, we have expanded the discussion of how the other study explain the asynchronous precipitation changes. See lines 323-333.

L284 If there was indeed a direct comparison with the classical theory (see earlier comment) this statement would be easier to follow.

A: As suggested, we have discussed the difference between our hypothesis and the classical theory. See lines 299-312. Also, this sentence has been revised to clearly compare the difference of classical theory and our hypothesis.

*Line 337-340: "Compared to the classical theory, which highlights the role of summer insolation in driving a synchronous ITCZ migration, we hypothesize an asynchronous nature of low-latitude precipitation optimums following the shifting perihelion. This offers a more plausible explanation for the observed asynchronous pattern of low-latitude precipitation's response to precessional forcing."*

Figure 5: Is this data also available? This would be valuable to test the hypothesis for other records. Moreover, could the y-axis be complemented with secondary floating/ relative timescale? This would make a comparison with comparison with proxy records from different latitudes easier.

A: The model simulated precipitation data is available from https://zenodo.org/records/13681177. Our results do not represent reality, since it is precessional cycle simulated by idealized model setup. However, comparison can be done by converting the age of proxies into precessional phases, by using Berger (1978) program as in Figure 2. For convenient, we also upload the calculated perihelion latitude at https://zenodo.org/records/14879991.

**References**

Baker, A., Hartmann, A., Duan, W., Hankin, S., Comas-Bru, L., Cuthbert, M. O., ... & Werner, M. (2019). Global analysis reveals climatic controls on the oxygen isotope composition of cave drip water. Nature Communications, 10(1), 2984.

Bischoff, T., Schneider, T., & Meckler, A. N. (2017). A conceptual model for the response of tropical rainfall to orbital variations. Journal of Climate, 30(20), 8375-8391.

Kaushal, N., Lechleitner, F. A., Wilhelm, M., Azennoud, K., Bühler, J. C., ... and SISAL Working Group members: SISALv3: a global speleothem stable isotope and trace element database, Earth Syst. Sci. Data, 16, 1933–1963, https://doi.org/10.5194/essd-16-1933-2024, 2024.

Parker, S. E., Harrison, S. P., Comas-Bru, L., Kaushal, N., LeGrande, A. N., and Werner, M.: A data–model approach to interpreting speleothem oxygen isotope records from monsoon regions, Clim. Past, 17, 1119–1138, https://doi.org/10.5194/cp-17-1119-2021, 2021.

Patterson, E., Skiba, V., Wolf, A., Griffiths, M., McGee, D., Bùi, T., ... & Johnson, K. (2024). Local hydroclimate alters interpretation of speleothem δ 18O records. Nature Communications, 15(1), 9064.

Singarayer, J.S., Valdes, P.J. & Roberts, W.H.G. Ocean dominated expansion and contraction of the late Quaternary tropical rainbelt. Sci Rep 7, 9382 (2017). https://doi.org/10.1038/s41598-017-09816-8

Treble, P.C., Baker, A., Abram, N.J. et al. Ubiquitous karst hydrological control on speleothem oxygen isotope variability in a global study. Commun Earth Environ 3, 29 (2022). https://doi.org/10.1038/s43247-022-00347-3

Windler, G., Tierney, J. E., & Anchukaitis, K. J. (2021). Glacial-interglacial shifts dominate tropical Indo-Pacific hydroclimate during the late Pleistocene. Geophysical Research Letters, 48, e2021GL093339. https://doi.org/10.1029/2021GL093339

Wu, Y., Warken, S., Frank, N., Mielke, A., Chen, C. J., Li, J. Y., & Li, T. Y. (2023). Northern Hemisphere summer insolation and ice volume driven variations in hydrological environment in southwest China. Geophysical Research Letters, 50(23), e2023GL105664.

A: These references were adopted and cited in different parts of our revision.

**Reviewer #3**

I think the message of this manuscript – that the timing of perihelion is a stronger determinant of precession-induced tropical precipitation peaks, rather than peak hemispheric insolation – is a worthwhile message.   The concept of the 'latitude of perihelion' that they introduce seems to be useful and I like how it intuitively explains the half-precession cycle seen in equatorial records.   Overall, I am in favor of this study.   My main concern is with the quality of the manuscript both in terms of the writing and presentation, and with the analysis, and I detail my concerns in the major comments.   With the latter, I find that their not using a fixed angle calendar to be highly problematic.   I think both can be (and should be) improved in keeping with the high standards of the journal.

A: Thank you for your valuable time to reviewing our manuscript. As suggested, we have made a calendar correction on the model results. After that, the results seem better to fit our hypothesis. Also, the writing and presentation have improved. Please checked the detailed response as below.

Major comments

1.   Writing and presentation:

I sense that the manuscript is formatted for a short-form journal but it is submitted as a research article in CP.   As such, there is a misalignment of styles and the manuscript is missing details which should be included in the main text.   I've mentioned some of the details in specific comments (e.g. a proper conclusion should be included).   Also, there are several figures in the supplementary which I think should be promoted into the main text (S4 in particular).

A: The very first draft was submitted to Nature on 23. Sep 2023. As suggested, in the revision, we have reformulated the paper to include a conclusion. Figure S2 and S4 have been moved to the main text.

Also, the writing needs to be improved and proofread for English grammar.   I've included several instances in the specific comments, but this is not a complete list and the entire manuscript should be edited.

A: We have gone through the paper several times to check the grammar errors. Hopeful, the revision is satisfied.

1.   Analysis

My main concern is that the authors do not correct for the 'calendar effect' in their simulations (lines 92-96).   Given that they use a high eccentricity (e=0.058), not doing a calendar correction to a fixed angle calendar can lead to substantial errors in comparison between your run cases, especially since you are examining specific

months (e.g. figure 5).   For example, Table 1 of Pollard and Reusch (2002) shows that for the 126ka BP (where e ~ 0.04) the offset between the Gregorian month and the fixed angle month is zero for April, and 13 days for October.   For a case where the longitude of perihelion is 180 degrees from the 126ka case, the offsets would be roughly reversed.   The offsets would be more extreme for your case (e=0.058), perhaps by a month.   It means that if you are comparing (say) October precipitation (as defined by the Gregorian calendar) at different longitudes of perihelion, you aren't doing a proper comparison of the monthly rainfall in terms of their relative positions in the Earth's orbit around the Sun.   This would be especially pertinent to your analysis in figure 5.

A: Following your suggestion, we have checked the calendar effect on the summer solstice and autumn equinox, the maximum date shift can be 19 days and 27 days, respectively. To improve this, we have applied a calendar correction to transfer our model results from Gregorian calendar to a fixed anglar canlendar. Detailed information can be found in Line 104-114. After the correction, the results in Figure 5. have improved. The precipitation maximum fit better with the perihelion latitude.

*Lines 104-114: "The definition of seasonality, which is influenced by slow variations in the Earth's orbit, plays a key role in determining the calculated seasonal cycle of the climate. Application of the Gregorian calendar where the lengths of the months and seasons are fixed results in a drift in the occurrence date of different seasons. Especially, the applied high eccentricity (0.058) leads to a shift in the date of the autumn equinox by up to 27 days within our simulations. This may lead artificial biases when comparing monthly temperature and precipitation across different simulations with different precessional phases (Kutzbach and Gallimore, 1988; Joussaume and Braconnot, 1997). In contrast to the "fix-day" Gregorian calendar widely used today, the angular calendar calculates the lengths of the months and seasons according to a fixed angle along the Earth's orbit. When comparing simulation results for different orbital configurations, it is essential to use the angular calendar to ensure that the data for comparison are from the same position along the Earth's orbit (Joussaume and Braconnot, 1997; Pollard and Reusch, 2002). To address this, we applied a calendar correction on our model results (temperature, precipitation, insolation) by changing the monthly mean data from the Gregorian calendar to an angular calendar. Detailed methodology can be found in Shi et al. (2022)."*

There are several places where I felt that a more rigorous analysis could be done, or some points are claimed without sufficient evidence.   I've detailed them in the specific comments.

Specific comments

Line 7 – argue, not argued

A: This sentence has been removed.

Line 8 – unclear what 'shifting perihelion' means. Suggest: 'shifting in the calendar timing of perihelion'

A: To explain what a shifting perihelion means, we have included two more sentences to explain what it means.

*Line 7-11: "In this study, we performed theoretical analysis, climate simulations, and geological records to hypothesize that the low-latitude hydrological cycle is paced by shifting perihelion rather than the hemispheric summer insolation. More specifically, precession of the Earth's rotation axis shifts the season and latitude of perihelion. Here, the latitude of perihelion is introduced as the latitude of Earth's subsolar point during perihelion, which is the location where the most intense solar radiation is concentrated. "*

Line 9 – unclear what 'occurrence season' means. Can you rewrite? Also, you haven't defined the latitude of perihelion yet in the abstract, so this is also unclear

A: As suggested, we have rewrite these sentences as above.

Line 15 – low latitudes

A: Corrected.

Line 29 – Suggest rewriting as "Low latitude precipitation seasonality primarily comes from the seasonal migration of the Inter-Tropical Convergence Zone, which in turn is governed by the interhemispheric contrast in insolation". This leads to a better tie-in to the sentence following.

A: The sentence is rewritten in the revision.

*"Low-latitude precipitation primarily comes from the seasonal north-south migration of the Inter-Tropical Convergence Zone (ITCZ), which follows the march of the Earth's thermal equator (Fig. 1)"*

Line 39 – the actual result is that the peak EASM at different geographical locations were asynchronous **with each other**, suggesting that the precipitation optimum doesn't occur all at the same time.

A: Thanks for clarification. We have modified this sentence according to your suggestion.

Line 50 hypothesize

A: Corrected.

Line 52 presence of the ice sheet. Also by 'development of vegetation', do you mean alteration of vegetation?

A: Corrected. The 'development of vegetation' has been deleted.

Line 55-56 Rewrite as "This raises question of whether terrestrial precipitation follows (or not) changes in hemispheric summer insolation "

A: Thanks for your sentences. We have adopted in the revision.

Line 57 instead of 'occurrence time', how about ' calendar timing'

A: Changed.

Line 60 – what do you mean by 'time-transgressive'?

A: To clearly explain what is time-transgressive, we have expanded this sentence to "…suggested that insolation in different months may contribute to a time-transgressive pattern of East Asian monsoon optimums, with earlier occurrence in the southern China and later occurrence in the northern area."

Line 64-65 Rewrite as "In this study, we use theoretical analysis, climate model simulations and analysis of geological records, to argue that the low-latitude hydrological cycle is paced by the calendar timing of perihelion, rather than variation in hemispheric summer insolation. "

A: Thank you for your suggestion. Calendar is an artificial concept. There are many different calendars in the world. If we use anomalistic year as our calendar, there will be no calendar timing shift of perihelion. What we want to emphasize is the latitude of perihelion. The sentence has been revised as follow:

*"In this study, we present theoretical analysis, climate model simulations, and synthesis of geologic records from different latitudes to hypothesize that the precession-driven low-latitude hydrological cycle is regulated by shifting perihelion across different seasons and latitudes. Consequently, the precipitation optimums at different latitudes occurs in a naturally asynchronous manner."*

Line 88: why 0.058 specifically?   Is it the maximum Earth's orbital eccentricity can attain?

A: It is almost the highest eccentricity during the quaternary. We have explained this in Line 100.

Section 2.3.   Can you provide more details on the EOF method, specifically did you account for the difference in areas for each gridpoint?   Is there a reference for the EOF method?

A: We have provided a reference for EOF. The difference in areas for each grid point are not take into accounted as in many other studies. However, as we only applied EOF analysis to the low-latitude area, where the grid difference is small.

Line 128-130. This isn't quite right.   The time it takes for successive perihelions is the anomalistic year, 365.2598 days.   This compares to the tropical year (successive vernal equinoxes) of 365.2422 days, the latter which the Gregorian calendar is based.   The difference between the two is around 25.1 minutes.   I used ChatGPT to get these numbers, so please verify.

A: Thank you for pointing out this. We misunderstand the gap between the tropical year and Sidereal year. You are right, two successive perihelion takes around 365.259636 days. The difference to tropical year is around 25.12 minutes. I have verified this information on https://www.britannica.com/science/year#ref120291.

Anomalistic Year: 365.259636

Tropical Year:      365.242190

Sidereal Year:      365.256363

Line 132.   As mentioned before, instead of occurrence season, I suggest 'calendar timing'

A: As suggested, we have revised this sentence to

"*By changing the orientation of the Earth's rotation axis, precession gradually delays the calendar timing of perihelion by around 25.1 minutes per year.*"

Line 142.   While you zero out obliquity, there may be some other boundary conditions that have a seasonal cycle in them (e.g. vegetation).   Can you elaborate, and would they affect your conclusions if so?

A: Our model has adopted a dynamic vegetation. So the vegetation is a response to Earth's orbit, and precipitation and temperature changes, not a forcing factor. We have introduced the dynamic vegetation in the revision.

*Line 81-82: "In our experiments, we implement a dynamic vegetation, which dynamically alters the vegetation coverage and the surface albedo in response to climate changes."*

Figure S2.   Is the average over all land, or just land in the tropics?   What about the ocean?   Since you are looking mainly at tropical rainfall and the MSE diagnostic works best over the tropics, I suggest doing tropical averages.

A: Previously, we do have used all the global land. As suggested, in the revision, we only apply the analysis on the tropical area (30S-30N). The corresponding figure have been updated.

Lines 146-148 "Due to different thermal inertia between the land and the ocean, the atmospheric heating over land is much stronger than that over the ocean. This leads to faster increase in moist static energy over the land than the ocean (Fig. S2)."   Its unclear to me whether these statements are correct.   First, while the land MSE does increase faster than ocean, it is not 'much' faster.   Also, without looking at the individual terms, how do you know that the increase in MSE is primarily due to surface temperature?   Could the moisture be a contributing or even dominant factor?

A: We have removed "much". The MSE have considered both temperature and moisture. Therefore, we do not clearly separate them. When there is precipitation, the surface temperature gets cooled.

Line 179: tropical convection, not tropical convergence zone

A: Modified.

Line 184: instead of 'Moving perihelion', use 'The shift in the calendar timing of perihelion'

A: We have revised this sentence to Line 213: "*In a precessional cycle, the perihelion shifts toward different seasons and perihelion latitudes.*"

Line 185: use 'tropical latitudes' rather than 'tropical area'

A: Modified.

Line 211: use 'test' rather than 'validate'.   Hypotheses can only be tested, not validated.

A: Modified.

Figure 5: one of the labels is incorrect.   Aug, not Agu

A: Thank you for your careful read. The error has been corrected.

Line 220-222: While I see the China precipitation peak at 90 degrees longitude of perihelion, it stays high until after 180 degrees longitude of perihelion.   This is at odds with the relative insolation peak which occurs over a narrower window of longitude of perihelion. On the other hand, Brazil rainfall peaks appear to be shorter-lived and correspond to their isolation peaks.   Please explain the discrepancy?

A: Very insightful question. Our answer is that we don't exactly know the reason. The long-term precipitation optimum found in China primarily exist during the last interglacial. We expect that relatively high CO2 may contribute to maintaining a strong East Asian summer monsoon. This long-term precipitation optimum disappears when the CO2 reduces, such as the case in 82.8 ka. We have included some discussion on this point.

*Line 264-271: "Regionally, there are not perfect agreements between the seasonal insolation variations and speleothem records. For example, the Chinese proxies peak at 90° longitude of perihelion around 127.3 ka. It stays high until after 121.7 ka. This long-term precipitation optimum was likely maintained by relatively high CO2 during the last interglacial period. Moreover, millennial scale abrupt climate changes originating from the North Atlantic also play a role in shaping the precipitation across the low latitudes (Wang et al., 2008; Chiang and Friedman, 2012). This is evident in the abrupt jumps in $\delta_{18}O$ signals from all different latitudes (Fig. 2). In addition, we noticed that the manifestation of October insolation in the half precessional signals in the Malaysia record is relatively stronger than that of the insolation in March. This is likely attributed to a stronger boreal autumn rain than the boreal spring rain in Malaysia (Fig. S1)."*

Lines 228-229 "We find maximum precipitation signals in the Malaysia speleothem record (Fig. 1c, red line) around 110.9 and 88.5 ka, corresponding to two maxima in March insolation. " I don't see what is claimed in in figure 1c. Rather, the maxima occur aournd 121.7ka and 99ka, and I see relative minima when the descending March insolation intersects with the rising October insolation around 105.5ka and 82.8ka. Please explain.

A: Indeed, the maximum at 121.7ka and 99 ka are robust, corresponding the rainy season in boreal autumn. And there are also two maxima around 110.9 and 88.5 representing precipitation maximum in boreal spring. For the minima, they are found around 116.2, 105.5ka and 82.8ka. They correspond to precession phase of 90 and 270, when the insolation in both rainy seasons is relatively weak. We have included this detailed information in our revision.

*Line 260-263: "For the East Asian and the South American summer monsoons, they reach their minima during the seasonal insolation minima. In contrast, at the equator, the minimum precipitation occurs not during seasonal insolation minima but during the time when insolation is relatively weak in both rainy seasons, i.e., at the precession phases of 90 and 270 (Fig. 6c), approximately 116.2, 105.5, and 82.8 ka (Fig. 2c)."*

Line 238: 'An earlier study....'

A: Modified.

Section 7: This section is mainly a discussion and there isn't a summary of the study.   Please include a proper summary of the study in this section, before going into discussion.

A: As suggested, we have included a summary at the beginning of discussion section.

*Line 283-289: "In this study, we investigate the dynamics of asynchronous evolution of low-latitude precipitation under precession forcing, a long-standing conundrum in paleoclimate research. We hypothesize that the precessional-scale low-latitude hydrological cycle is paced by shifting perihelion rather than the hemispheric summer insolation. Using two sets of idealized climate model simulations, we showed that whenever and wherever perihelion occurs, the tropical terrestrial precipitation peaks in the corresponding perihelion season and latitudinal band. Under this new framework, the low-latitude precipitation naturally follows distinct rhythms. Speleothem proxies from three typical latitudes were included to test our hypothesis, which shows that the regional precipitation optimum matches the meridional migration of perihelion latitude."*

Line 245: Instead of "Many transient simulations", how about "Previous studies have used transient simulations to…"

 A: Modified.

References

Pollard, D. and Reusch, D.B., 2002. A calendar conversion method for monthly mean paleoclimate model output with orbital forcing. *Journal of Geophysical Research: Atmospheres*, *107*(D22), pp.ACL-3.
**Citation**: https://doi.org/10.5194/egusphere-2024-2778-RC2

A: This reference is cited in the introduction of calendar correction.

---

## Author Response (AR2)

Dear Editor and Reviewers,

Thank you again for your time and effort for evaluating and improving our manuscript. Following the reviewers' suggestion, we have made necessary changes and point-by-point answers.

Best,
Hu Yang on behalf of all coauthors.

**Anonymous referee #1**

In my view the revised version of the manuscript has significantly improved, and the authors have adequately handled most of the revision requests. I have only a few, very minor issues that should be tackled before acceptance:

A: Thank you for your encouragement and valuable time.

- L8 (abstract): include "synthesis of (geological records)" (similar to what is written later on in the introduction.

A:Modified.

- I am not fully satisfied with the response to my comment to the statement in the new L245ff. As I have written in my previous review, this statement is not entirely true. In tropical settings, speleothem growth is often biased towards the winter season due to temperature related ventilation (e.g., Sekhon et al., 2021, Vieten et al., 2016, Voarintsoa et al., 2021, ...). BUT (and I think that is what the authors mean) - the d18O of the speleothems is usually a precipitation weighted-annual average, which is then biased towards the rainy season (which is often the summer). But this has not necessarily to do with the season of deposition, because usually, the drip water is a mix of rainwater over several months to years. This is important because the non-expert reader may misunderstand this statement as is, so please clarify this, e.g. by writing "Tropical speleothem d18O records typically reflect an amount-(or infiltration) weighted annual mean precipitation d18O, and are therefore typically biased towards the rainy season."

A:Thank you for your clarification and accurate words. We have replaced the sentences as suggested.

Line 253-254: "*Tropical speleothem δ18O records typically reflect an amount-(or infiltration) weighted annual mean precipitation δ18O, and are therefore typically biased towards the rainy season.*"

- In L317ff the authors briefly mention a potential impact of their result on the practice of astronomical tuning. I suggest to add 1-2 more sentences, that explain how significant this is, e.g., explicitly name a constellation, when and where such tuning may be significantly wrong, and by how much chronologies may be "off". This could include also a suggestion how astronomical tuning could be improved.

A: As suggested, we have extended our discussion on the astronomical tuning and provide solution for improving the tuning.

Line 324-331: *"Astronomical tuning is widely used to establish the chronology of paleo proxies. By doing this, the phasing of proxies is artificially synchronized. However, our results indicate that the astronomically driven climate changes can naturally follow diverse rhythms. This questions the reliability of the astronomical tuning strategy. For example, absolutely dated proxies indicate an asynchronous onset and termination of Greening Sahara at different latitudes (Kuper and Kropelin, 2006; H.ly et al., 2014; Shanahan et al., 2015). Synchronizing the ages of proxies from different regions may lead to biases of a few millennia and introduce difficulties in understanding their dynamics. Based on our model simulations (Fig. 6), we suggest that astronomical tuning should target the insolation 1-2 months before the local rainy season, at least for precipitation-related proxies in low-latitudes."*

References

Sekhon, N., Novello, V. F., Cruz, F. W., Wortham, B. E., Ribeiro, T. G., & Breecker, D. O. (2021). Diurnal to seasonal ventilation in Brazilian caves. Global and Planetary Change, 197, 103378.

Vieten, R., Winter, A., Warken, S. F., Schröder-Ritzrau, A., Miller, T. E., & Scholz, D. (2016). Seasonal temperature variations controlling cave ventilation processes in Cueva Larga, Puerto Rico. International Journal of Speleology, 45(3), 7.

Voarintsoa, N. R. G., Ratovonanahary, A. L. A. J., Rakotovao, A. Z., & Bouillon, S. (2021). Understanding the linkage between regional climatology and cave geochemical parameters to calibrate speleothem proxies in Madagascar. Science of the Total Environment, 784, 147181.

**Anonymous referee #2**

I thank the authors for the comprehensive revision including the implementation of the fixed angle calendar. I only have a few minor clarification comments remaining. Otherwise, this is a nice study with a novel view on the interpretation of how precession alters tropical rainfall seasonality, and deserves to be published.

A: Thank you for your encouraging words and time.

1) Lines 140-147. The description of the EOF needs sufficient detail so that it can be reproducible. Specifically: what latitudes are you applying the EOF; and are you accounting for area weighting in the calculation? I assume that the first mode strongly dominates (as shown by the variance explained) so you don't need to worry about the robustness of EOF1.

A: As suggested, we have included the area (40S-40N) where the EOF analysis was applied. Area weighting is not applied.

Line 145-151: *"The Empirical Orthogonal Functions (EOF) analysis (Hannachi et al., 2007) is used to identify the spatial and temporal characteristics of terrestrial precipitation at low latitudes. EOF analysis is widely applied in Earth science. It is generally used to simplify a spatial-temporal data set by converting it into spatial patterns of variability and temporal evolution of these patterns. For the idealised Earth system experiment without tilted Earth rotation axis, we applied EOF analysis on the climatology monthly convective precipitation over land between 40oS-40oN (Fig. 4). For the 24 simulations recovering a precessional cycle, we applied EOF analysis on the individual monthly convective precipitation over land. For example, December precipitation in the 24 simulations is selected and applied a EOF analysis to generate Fig. 5a. Area weighting is not applied in the calculation."*

2) Figure 5 and associated caption. I have a few suggestions for improvement:

i) The pink dot is hard to see, please use a different color/shape?

A: As suggested, we have replaced the pink dot to red arrow to better illustrate the location and timing of the perihelion.

ii) Also, what is the area taken in the averaging of the solar radiation and temperature for the curves in the left hand panels? Please specify in the caption.

A: We have selected the region show on the map, i.e., 40S-40N to calculate the temperature and solar radiation. The corresponding information is added in the new caption.

iii) For the caption: 'climatological precipitation' not 'climatology precipitation'

A: Corrected.

iv) Also, when you say "the plotted solar radiation is for an earlier month', do you mean one month before, or are you picking the best month that fits with the other curves? If the latter, how would you justify it give that the delay in the precipitation response from the insolation is a physical process that should be more or less the same regardless of the month

A: The plotted solar radiation is from one month before, not artificially selected. For example, for the subpanel of February precipitation, the insolation in January is plotted; for the panel of August precipitation, the solar radiation in July is plotted. There is logical reason for this. As

the solar radiation at one month early will determine the temperature and precipitation in the next month.